# Fundamentals of vaping-associated pulmonary injury leading to severe respiratory distress

Carolina Esquer*†, Oscar Echeagaray*†, Fareheh Firouzi*, Clarissa Savko*, Grant Shain*, Pria Bose, Abigail Rieder, Sophie Rokaw, Andrea Witon-Paulo, Natalie Gude, Mark A Sussman

**Vaping of flavored liquids has been touted as safe alternative to traditional cigarette smoking with decreased health risks. The popularity of vaping has dramatically increased over the last decade, particularly among teenagers who incorporate vaping into their daily life as a social activity. Despite widespread and increasing adoption of vaping among young adults, there is little information on long-term consequences of vaping and potential health risks. This study demonstrates vaping-induced pulmonary injury using commercial JUUL pens with flavored vape juice using an inhalation exposure murine model. Profound pathological changes to upper airway, lung tissue architecture, and cellular structure are evident within 9 wk of exposure. Marked histologic changes include increased parenchyma tissue density, cellular infiltrates proximal to airway passages, alveolar rarefaction, increased collagen deposition, and bronchial thickening with elastin fiber disruption. Transcriptional reprogramming includes significant changes to gene families coding for xenobiotic response, glycerolipid metabolic processes, and oxidative stress. Cardiac systemic output is moderately but significantly impaired with pulmonary side ventricular chamber enlargement. This vaping-induced pulmonary injury model demonstrates mechanistic underpinnings of vaping-related pathologic injury.**

## Introduction

Relative merit of vaping for "harm reduction" intervention that transitions smokers away from combustible cigarettes remains under attack because of widespread adoption of vaping as a social activity and lifestyle choice by "never smokers," particularly adolescents [1, 2, 3]. The electronic vaping-associated lung injury (EVALI) outbreak of 2019 serves as a sobering demonstration of potential dangers resulting from uninformed experimentation with vape juice composition [4, 5, 6]. In comparison, commercially sold vape juices and prefilled disposable devices typically do not provoke acute lung injury and respiratory distress as pointed out by vaping advocacy groups [7, 8, 9]. However, vaping-associated pulmonary injury (VAPI) in a minority of users is a well-documented and growing concern in the clinical setting with multiple independent reports of severe respiratory illness including acute respiratory distress syndrome including potential pulmonary circulation impairment [10, 11, 12, 13, 14, 15, 16, 17]. Life-threatening consequences of VAPI emphasized by high profile media coverage over the last couple of years have raised public awareness of danger linked to vaping [18, 19, 20, 21] with increased calls for regulatory oversight and expanded research [22, 23]. Fundamental unresolved issues using commercial vaping products from reputable sources include (1) how do individual-specific biological factors influence susceptibility to VAPI; (2) what is the underlying pathogenesis of VAPI leading to respiratory distress; and (3) how does progression of VAPI precipitate pulmonary circuit failure? Immediacy of the problem in society, recency of modern electronic vaping, and the rapid evolution of vaping technology necessitate development of an innovative platform to study biological processes of VAPI.

Studies of VAPI often target specific biological processes to assess consequences for phenotypic or functional impact in cells, tissues, or animals. Assessment of vape fluid exposure in vitro is typically focused upon a particular cell type such as epithelial or vascular cells [24, 25, 26, 27, 28]. Alternatively, in vitro studies of mixed cell cultures offer insight into disruption of structural interactions in "organoid" settings [29]. Primary culture of explanted tissue is another option for assessing consequences of vape juice exposure [30, 31]. Observation of in vitro system is advantageous for several reasons including simplifying, focusing, and increasing throughput with targeted analyses of select cell or tissue types, but extrapolating findings to the in vivo setting remain problematic without corresponding animal studies. Experimentation conducted using animal models shows varying outcomes after vape juice exposure ranging from negligible to substantial [24, 32, 33, 34]. Inconsistent findings are undoubtedly due, at least in part, to variability in exposure protocols. Relatively few studies provide

San Diego State University Integrated Regenerative Research Institute and Biology Department, San Diego State University, San Diego, CA, USA

Correspondence: heartman4ever@icloud.com
*Carolina Esquer, Oscar Echeagaray, Fareheh Firouzi, Clarissa Savko, and Grant Shain contributed equally to this work
†Carolina Esquer and Oscar Echeagaray are co-first authors

compelling rationales for experimental design or acknowledge inherent limitations of their approach (35). Critics of vaping research leverage issues of experimental interpretation and variation to challenge relevance of findings, leading to pitched debate between pro-versus anti-vaping advocacy groups (36, 37). Establishing reasonable approximations of human vaping behavior as well as clinical manifestations of VAPI is essential and desperately needed to advance research and promote consensus among all stakeholders.

Quintessential combined features of VAPI are most appropriately recapitulated in a mammalian inhalation exposure model. Capturing the numerous variables of human vaping activity such as (but not limited to) frequency of use, vaping inhalation topography, device type, and vape juice formulation create a daunting challenge for implementing a "typical" exposure protocol. Nevertheless, topography and demographics of vaping have been studied (38, 39, 40, 41), whereas user preferences for devices and juices are a moving target subject to social and marketing influences (42). Presently, JUUL remains the most popular e-cigarette device with 51.6% of the reusable market share in April 2021 and 49.7% of the entire e-cigarette market (43, 44, 45). Vape juice preferences trend toward fruit flavors, often with the addition of menthol to provide a cooling sensation (46, 47). Considerations such as these should be collectively incorporated into experimental design if the intent is to target human behaviors and choices.

Rapid evolution of the vaping industry coupled with unpredictable trending preferences of end users can have devastating consequences such as the EVALI outbreak, which was quickly correlated to ill-conceived modifications in "Dank Vape" juices (48, 49). However, severe respiratory distress syndromes linked to VAPI are increasingly seen in the clinical setting involving vapers using commercially sourced devices and vape fluid (15, 50, 51, 52, 53, 54). Evolution and onset of VAPI in otherwise healthy individuals remains poorly understood, especially on cellular and molecular levels. Thus, the emerging clinical syndrome of VAPI forms the basis of this report wherein an experimental model of inhalation exposure was developed and characterized. Findings presented here document, to our knowledge, the first VAPI inhalation exposure model instigated using popular products sourced entirely from retail markets revealing novel biological responses and pathogenic processes.

# Results

## Structural and morphometric alterations to lung parenchyma and airways following vape exposure

Tissue samples harvested following Week 9 of exposure comprised the excised respiratory tree consisting of trachea with lungs, with hearts concurrently processed as detailed in the Materials and Methods section. Termination of the study at Week 9 time point was chosen based upon longitudinal histologic observation of lung tissue exhibiting progressive pathological features of VAPI. Longitudinal time course histological assessments were performed on a small cohort of subjects beginning at Week 5 at 1-wk intervals for a small cohort (Fig S1). Overt behavioral or physiological stress was

not evident from daily routine monitoring of mouse subjects throughout the Week 9 time course. Lung sections were prepared in the coronal plane and visualized for basic structure by H&E stain (Fig 1). Normal non-vaped lung tissue shows typical features of

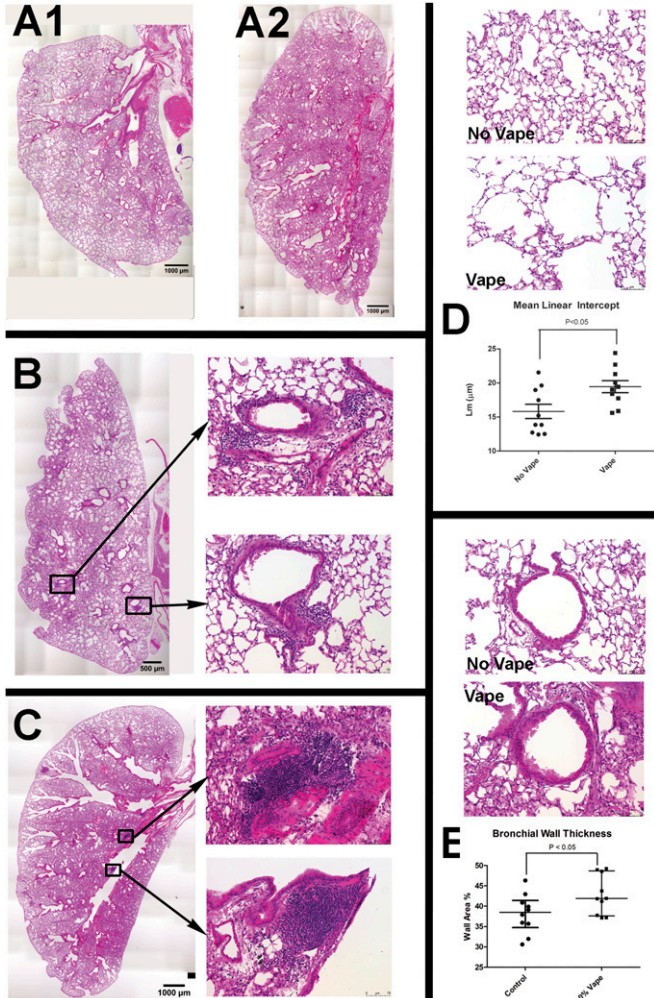

**Figure 1. Structural and morphometric alterations to lung parenchyma and airways after vape exposure.**
**(A1, A2)** Representative H&E images of entire lung section demonstrating overview of changes after vape exposure in normal (A1) and vaped (A2) lung. **(B, C)** Representative images of vape lungs showing cellular infiltrates proximal to airways (B, C).
**(B)** Overview of vape lung showing localization of cellular infiltrates surrounding bronchioles (B) with inset images at right showing boxed regions of cellular infiltrates surrounding bronchioles. **(C)** Overview of vape lung showing localization of cellular infiltrates at branching of primary bronchiole (C) with inset images at right showing boxed regions of cellular infiltrates at branching of primary bronchiole. **(D)** Lung parenchyma structure (D) in normal lung (No Vape) compared with vape lung shows alveolar rarefaction (Vape). **(D)** Average free distance between alveolar walls measured by mean linear intercept (Lm) shows significantly increased open space in vaped relative to normal lung (D, Mean Linear Intercept graph; $P < 0.05$). Each dot represents the average of all independent measurements from one mouse. The bars represent the median and interquartile range. **(E)** Bronchiole airway cross-section comparison (E) between normal (No Vape) versus vaped (Vape) lung tissue in sections. **(E)** Bronchiole wall thickness significantly increased in vape group (E, Bronchiole Wall Thickness graph; $P < 0.01$) measured as percentage of the difference between total and lumen area over total area. Each dot represents the average of all independent measurements from one mouse. The bars represent the median and interquartile range. N = 10 mice for each group.

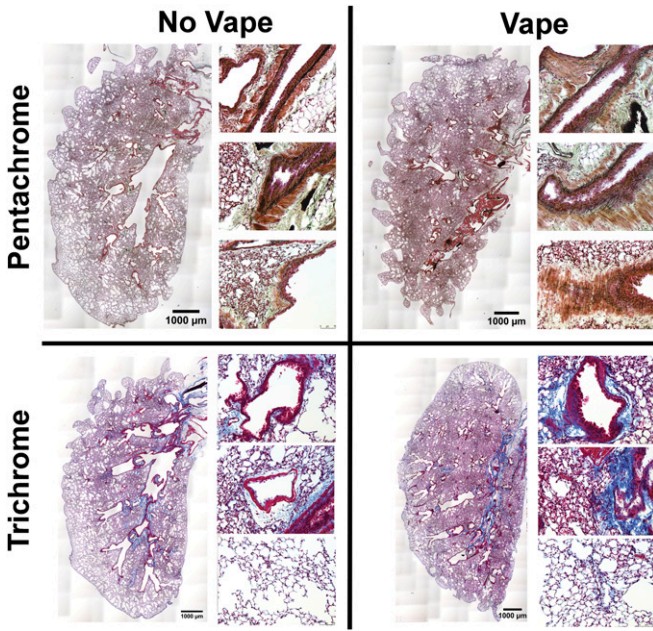

**Figure 2.  Pathologic disruption of structural organization in airway and lung tissue of vaped mice.**
Overview of lung sections from No Vape (left) and Vape (right) stained with either Movat's Pentachrome (top row) or Masson's Trichrome (bottom row). Pentachrome stained magnified images at right of each overview show elastic fiber disorganization lining the airways in Vape samples (elastic fibers and nuclei [black]; muscle [red]; fibrin [bright red]; collagen and reticular fibers [yellow]; and ground substance and mucin [blue]). Trichrome-stained magnified images at right of each overview show increased collagen deposition surrounding bronchioles (top), vasculature (middle), and parenchyma (bottom) in vape lungs (collagen (blue); muscle fibers (red); nuclei (black).

larger bronchi proximal to primary branch points becoming smaller toward peripheral regions, scattered vasculature of various sizes, and parenchyma with uniformly distributed alveolar areas (Fig 1, A1). In comparison, vaped lung shows increased staining intensity proximal to bronchi in central regions and remodeling of lung parenchyma (Fig 1, A2), suggesting pathological changes in bronchiolar and vascular structure. Closer visualization of bronchioles from vaped lung sections reveals wall thickening, deterioration of smooth muscle organization, and increases in cellular infiltrates (Fig 1B). Large accumulations of cellular infiltrate are particularly prominent near vasculature as clusters of numerous cells concentrate around injured vessels (Fig 1C). Widespread alveolar rarefaction in vaped lung tissue is present with significant losses in peripheral regions as revealed by alveolar space quantitation (Fig 1D). Adverse remodeling of thickened bronchiole walls was also significant (Fig 1E). Collectively, these findings demonstrate multiple pathological abnormalities consistent with VAPI and reveal the extent and nature of damage caused by inhalation exposure in the lungs.

## Pathologic disruption of structural organization in airway and lung tissue of vaped mice

Pursuant to findings using H&E (Fig 1), additional histological stains were used to further understanding of lung tissue composition.

Coronal lung sections from both non-vaped and vaped mice were stained with either Pentachrome or Trichrome dye mixtures to reveal distribution of elastin fibers or collagen, respectively (Fig 2). Normal non-vaped lung shows densely packed elastin fibers surrounding the periphery of airway passages originating in the central bronchi and continuing into the distal airspaces. In contrast, structural organization alteration of elastin fibers in the vaped lung is evident by loss of alignment, density, and orientation relative to the airway (Fig 2, top row). Damage to lung architecture is prevalent in vaped tissue sections with increased collagen deposition associated with bronchial airways, vessels, and alveolar spaces compared to the non-vaped control sample (Fig 2, bottom row). Estimation of collagen deposition indicates a 45.69% (1.5-fold) increase in the vaped lung compared to the non-vaped group (non-vaped covered 3.46% of area; vaped covered 5.04% of area) by methyl blue stain quantitation on images. These results provide further evidence of VAPI pathology and structural abnormalities induced by inhalation exposure of vape aerosol.

## Mucin accumulation in bronchial airways of vaped mice

Presence of mucopolysaccharides and glycoproteins in lung tissue sections was observed using periodic acid–Schiff (PAS) stain with detection visualized as a deep red Fuchsia color. PAS labeling intensity is increased in the lung tissue of vaped mice compared with non-vaped samples (Fig 3A). PAS stain was notably present in the epithelial lining of airway tracts in numerous large secretory vesicles. This initial finding was further confirmed by immunolabeling for mucin 5AC (Muc5AC), a major constituent glycoprotein of secretory mucus produced by goblet cells that protects airways from foreign pathogens (55, 56). Muc5AC immunolabeling is increased in bronchiolar goblet cells in the airway of vaped mice compared with non-vaped samples (Fig 3B). Cell membranes are visualized with immunolabeling for Epithelial-cadherin (ECAD), a cell adhesion molecule (57). Muc5AC immunolabeling shows distribution along the apical surface of epithelial lining as expected for a secretory protein. Elevation of mucin protein expression was confirmed by immunoblot analysis of protein homogenates prepared from the right lung lobe using antibodies to Muc5AC and Muc1 (Fig 3C). Protein levels for Muc5AC and Muc1 were significantly increased by 3.5 ± 0.36 and 2.3 ± 0.19-fold, respectively, as determined by quantitative analysis ($P < 0.001$). Taken together, these findings demonstrate increases in PAS stain and mucin accumulation in vaped lung samples consistent with expectations for VAPI and airway epithelial tract response to expunge xenobiotic agents.

## Pseudostratified columnar epithelium disruption in trachea of vaped mice

Upper airway epithelial lining play a critical role in the respiratory defense response to xenobiotic agents by trapping and clearing foreign pathogens (57, 58). Within the epithelial lining, ECAD serve an indispensable role in regulation of cell–cell adhesion as well as regulation of innate immunity (59). Thus, upper airway structure was evaluated with emphasis upon ECAD localization and protein expression level. Trachea sections from upper airway were also immunohistochemically labeled for the water-specific channel

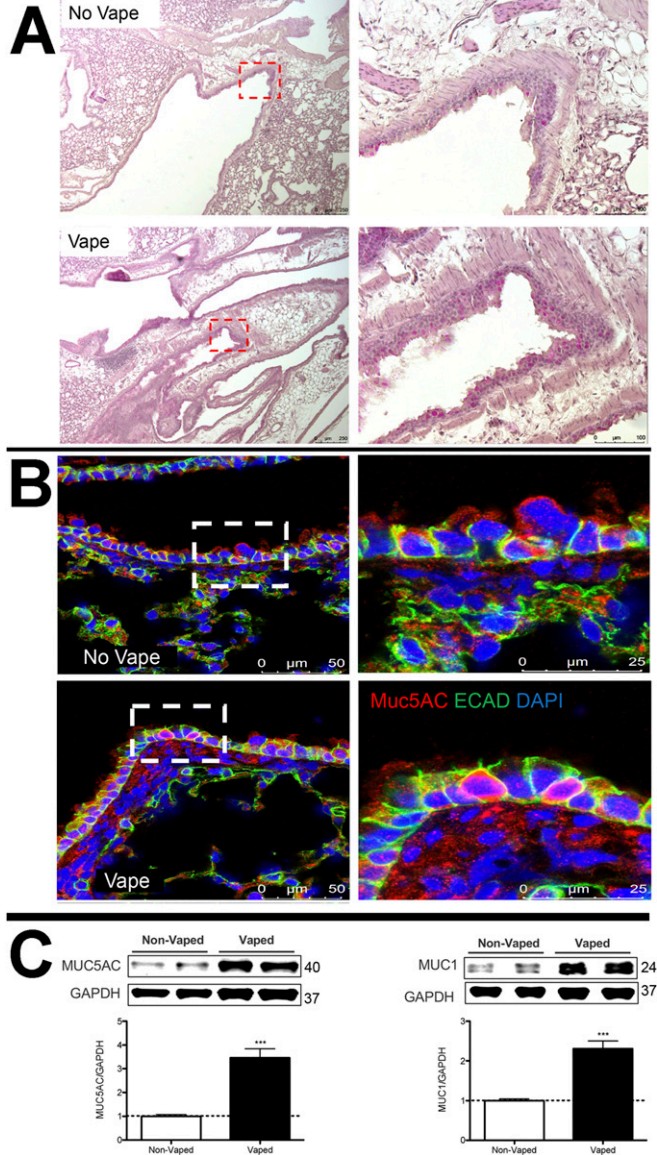

**Figure 3. Mucin accumulation in bronchial airways of vaped mice.**
**(A)** Periodic Acid Schiff stain of representative No Vape or Vape lung tissue sections demonstrating increased mucopolysaccharides (deep red color) in vaped samples. Inset images boxed by red dotted lines are shown to the right of corresponding section. **(B)** Increased Mucin 5AC in goblet cells of lower distal airway near the parenchyma by confocal immunomicroscopy. All images are representative of Mucin 5AC (red), E-cadherin (cell adhesion molecule; green), and DAPI (nuclei; blue). **(C)** Mucin 5AC and Mucin 1 protein levels are significantly increased (***$P$ < 0.001) in tissue samples prepared from lungs of vaped mice (Vaped) relative to normal lung samples (Non-Vaped). Corresponding quantitation of immunoblots is shown under each representative blot. GAPDH is used as loading control. Independent replicate blots of n = 12 for MUC5AC and MUC1 using four non-vaped control mice and 17 vaped mice. Error bars represent SEM.
Source data are available for this figure.

Aquaporin 5 (Aq5) (60), basal cells expressing cytokeratin 5 (Krt5), ciliated cells expressing $\alpha$-tubulin, and the cytoskeletal signaling molecule tetraspanin (61, 62, 63). Deterioration of upper airway pseudostratified epithelial organization is evident in samples from vaped mice versus non-vaped controls (Fig 4A and B). Normal

columnar structure remodels into a multilayered cellular sheet with dysmorphic features varying from collapsed (Fig 4A) to hyperplastic (Fig 4B). Basal cell localization along the basement membrane shows irregularities and Aq5 immunoreactivity is diminished in sections from vaped mice relative to non-vaped controls. Notably, ECAD immunoreactivity is enhanced in vaped sections versus non-vaped controls (Fig 4A and B) prompting further evaluation by immunoblot analysis of the right lung lobe that demonstrates a significant 30% elevation of ECAD protein in the vaped lung ($P$ < 0.01; Fig 4C). In summary, these results reveal multiple alterations of proteins involved in structural and functional properties of the lung airway and concomitant loss of pseudostratified cellular architecture in trachea epithelium.

### Inflammatory activity increased in vaped mice

Presence of cellular infiltrates observed by H&E (Fig 1C) suggests the presence of an inflammatory response in the lung of vaped mice. Immunophenotyping was performed using antibody to CD11b (also known as Mac-1), an integrin primarily expressed on monocytes, macrophages, neutrophils, DCs, NK cells, and a subset of B and T cells (64, 65) as well as CD11c, a marker of DCs. CD11b and CD11c immunoreactivity is increased in tissue sections from lung parenchyma of vaped mice compared to non-vaped controls (Fig 5A and B). Comparison of cellular density for CD11b and CD11c was determined by counting of cells in sections from two non-vaped and four vaped mouse samples. Four images were taken per sample, each with an area of 1.32 mm$^2$ totaling 5.27 mm$^2$ imaged per sample. Cell count for CD11b was significantly increased by fourfold (8.875 ± 1.62 for no vape versus 35.69 ± 4.66 vape; $P$ = 0.001. Cell count for CD11c was significantly increased by 2.3-fold (23.25 ± 7.2 no vape versus 53.19 ± 5.257 vape; $P$ = 0.003). Additional evidence supporting enhanced inflammatory activity in vaped lung samples was provide by immunoblot analysis of lung parenchyma homogenates for immune cell markers. Expression of CD11b increased 2.5 ± 0.33-fold in vaped compared with non-vaped control samples, along with similar elevations of CD45 (2.0 ± 0.12-fold), CD206 (1.4 ± 0.07-fold), and CD11c (1.9 ± 0.16-fold) (Fig 5C and D). Inflammatory cytokine expression was also elevated in vaped versus non-vaped lung tissue homogenates including IL-6 (2.1 ± 0.14-fold), IL-1 (1.9 ± 0.07-fold), and high mobility group box protein-1 (HMGB1; 1.4 ± 0.04-fold). Last, elevation of the extracellular matrix protein fibronectin (2.5 ± 0.20-fold) is indicative of inflammation and possibly tissue repair after bronchopulmonary injury (66, 67). Overall, these results indicate potentiation of inflammation consistent with VAPI in the lungs of vaped mice.

### Transcriptome profiling of alterations induced by vape inhalation exposure

Spatial transcriptional analysis demonstrates biological effects consistent with observations from microscopy and immunoblot analyses (Figs 1–5). Transcriptional data were obtained from cryosections of upper pulmonary branches from non-vaped (NV) and vaped (V) mice (two males and two females) using the Visium platform (10× Genomics) and aligned to the murine transcriptome (Fig 6A). All samples displayed comparable unique molecular

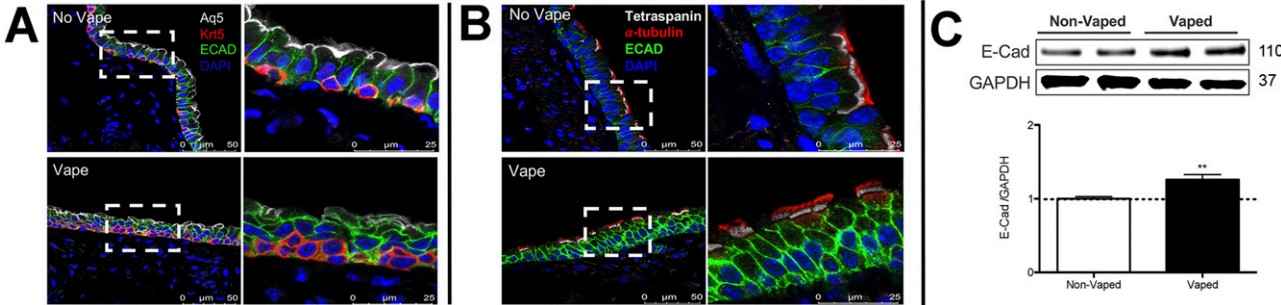

**Figure 4. Pseudostratified columnar epithelium disruption in trachea of vaped mice.**
Pseudostratified cell architecture in normal trachea epithelial layer (No Vape) is disorganized by vaping exposure (Vape). **(A)** Deterioration of Basal Cell (red; Krt5) distribution, diminished Aquaporin 5 channel immunoreactivity (white; Aq5) in surface epithelium, and increased immunolabeling of E-Cadherin (green; E-Cad) in vaped sample relative to No Vape control. **(B)**. Ciliated epithelium (red; α-tubulin) and cytoskeletal tetraspanin (white) marks apical epithelial boundary above disorganized cell layers in Vaped versus No Vape controls revealed by E-Cad (green) and nuclei (blue; DAPI). All images are representative and taken in superior trachea. **(C)** E-Cadherin (E-Cad) protein levels are significantly increased (**P < 0.01) in tissue samples prepared from lungs of vaped mice (Vaped) relative to normal lung samples (Non-Vaped). Corresponding quantitation of immunoblot is shown representative blot. GAPDH is used as loading control. Independent replicate blots of n = 6 using four non-vaped control mice and 17 vaped mice. Error bars represent SEM.
Source data are available for this figure.

identifier (UMI) and detected gene expression per spot with a slight differential between lung parenchyma and upper airway tissue (Fig S2). Unsupervised clustering revealed 14 clusters throughout all samples (Clusters 0–13), facilitating tracing of treatment-associated features within the pulmonary architecture (Fig 6B–D).

Spot distribution normalized to input revealed some clusters distributed in both NV and V samples (clusters 0, 3, 5, and 12; Fig 1E). Spots from clusters 2, 4, 8, and 10 were predominant in the NV group, whereas spots from clusters 1, 6, 7, 9, 11, and 13 were mostly present in V samples (Fig 6E). An expected factor driving transcriptional clustering was lung architecture as captured in the tissue section and number of transcriptionally similar spots (Fig 6F). Therefore, accurate interpretation of data spots required subset of lung parenchyma and upper airway for downstream analysis (parenchyma: clusters 0, 1, 2, 3, 5, 6, 7, 10, and 12; upper airway: clusters 4, 8, 9, 11, and 13; Figs 6D and S3). Spot mapping identified cell types in each cluster by score, with some spots mapping to multiple cell types (SF8 and Table S1).

Multiple cells from each lung section were captured on each spot (50 μm in diameter) of the gene expression slide, conferring the spots with hybrid transcriptome for spatial analysis. Cell type identification of hybrid gene expression spots was done by cross-referencing spatial transcriptome to the Mouse Cell Atlas (MCA) single-cell annotated database (1) (Fig 6A). Spot mapping identified cell types in each cluster by score, with some spots mapping to multiple cell types (Fig S4, part 1 and 2). Spot mapping mostly aligned to transcriptome annotated in the murine lung, but in some instances displaying transcriptional similarities to cell types from other tissues or developmental stages (Fig S5). Spot ratios per cluster were contextualized with the number of spot per cluster (Fig 6F and G). Endothelial and stromal cells were present throughout all clusters and lung regions, with some spots matching cell type subclassifications characterized by expression of particular markers (Figs 6G, S5, and S6). Lung parenchyma broadly displayed transcripts distribution of annotated adipocytes, AT1, AT2, B cells, club cells, DCs, and macrophages, whereas spots in the upper airway show higher transcript for ciliated and muscle cells (Fig S6).

Clusters 0, 3, 5, and 12 predominant in parenchyma in both NV and V samples captured endothelial, stromal, ciliated cells, club cells, muscle, lower proportion adipocyte, erythroblasts, and macrophages. However, clusters overall vaped samples showed higher spatial distribution of spots mapping to MCA annotated AT1, AT2, ciliated cells, and club cells (Fig S7), suggesting a shift on these populations in response to vaping treatment. Together, these findings define transcriptional changes on the lung architecture in cell type detection and spatial distribution in response to chronic vaping exposure.

### Transcriptional response to xenobiotic stimulus, endothelial apoptosis, and lipid catabolism in lung parenchyma

Transcriptional differences across treatments on the parenchyma tissue were revealed by differential expression analysis, with 364 differentially expressed genes (DEGs) identified on NV spots and 51 DEGs on V parenchyma tissue (Fig 7A). DEGs derived from differential expression analysis were used as input for gene ontology (GO) analysis. GO term analysis by biological processes revealed an enrichment of various ontologies associated with metabolic processes and cellular apoptosis in the V samples (Fig 7B). Gene targets belonging to GO term (GO: 0071466), cellular response to xenobiotic stimulus, showed consistently increased expression in V samples (Cyp2a5, Gsto1, Cyp2f2, Fmo2, Gsta3, and Cyp2b10; Fig 7D). Parenchyma of V samples showed higher spot number expressing Gsto1 and Cyp2f2 with increase distributions throughout the tissue (Fig 7C and D). Transcripts for processes of xenobiotic response and endothelial apoptosis were significantly increased in vaped compared to non-vaped samples (Fig 7E). Cidea, Acadvl, Hsd11b1, and Acer2 targets belong to the lipid catabolic process GO term (GO: 0016042) were up-regulated in the parenchyma of V samples. Consistent with exposure of vape fluid, Cidea, a gene associated to lipolysis, was overexpressed in vaped samples (Fig 7F). Additional GO terms with up-regulated targets upon vaping exposure included: drug catabolic process (GO: 0042737; Chil3, Cyp4b1, Hbb-bt, and Prdx6), glycerolipid metabolic process (GO: 0046486; Gpd1, Thrsp, Agpat2,

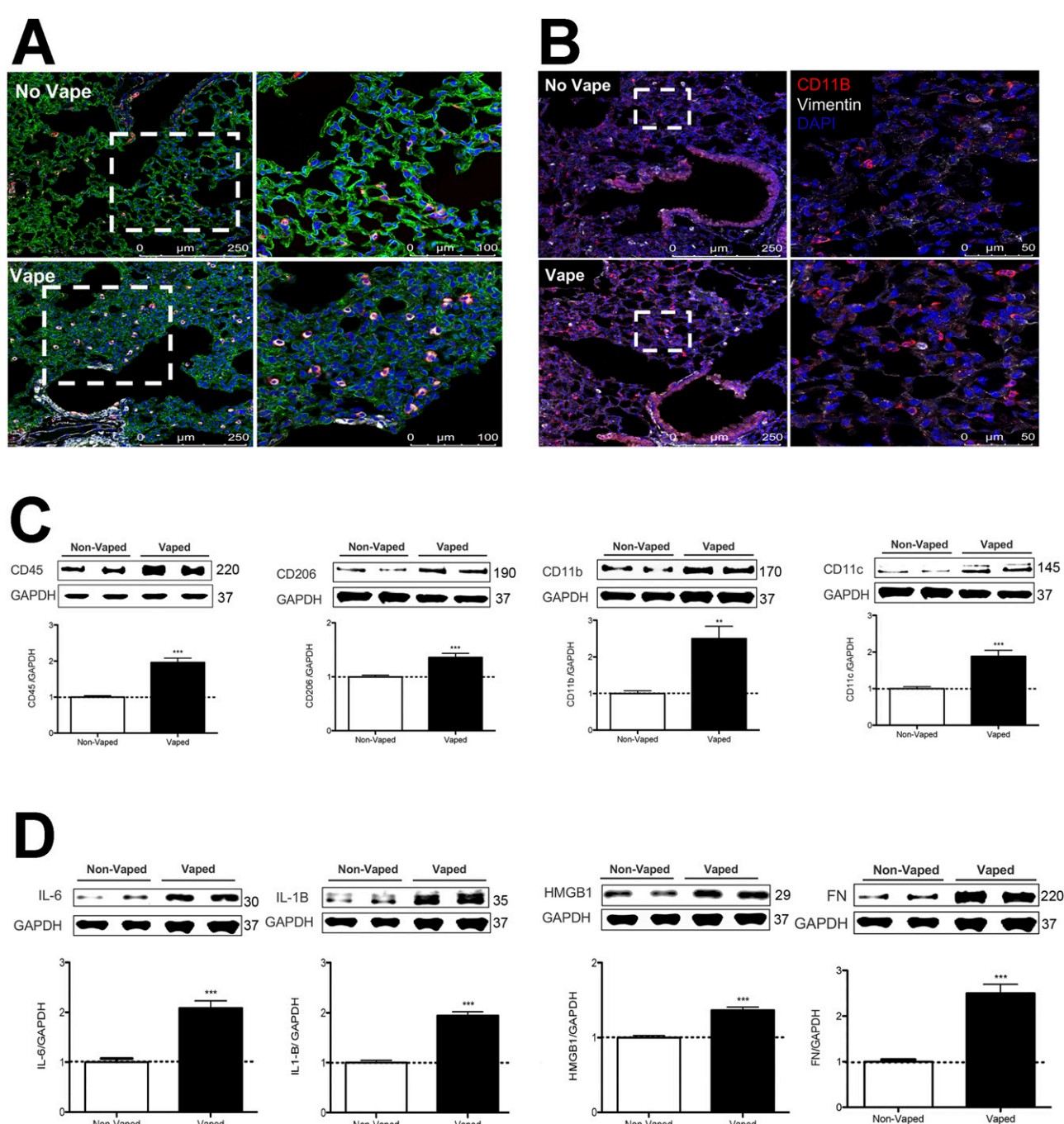

**Figure 5. Inflammatory activity increased in vaped mice.**
**(A)** Immune cellular infiltration increased in vaped (Vape) relative to normal (No Vape) lung tissue sections revealed by immunoreactivity for CD11c (innate immune cells; red), vimentin (intermediate filaments; white), and Receptor for Advanced Glycation End products (RAGE; alveolar type-1 cells; green) with nuclei label (blue; DAPI).
**(A, B)** Comparable immunolabeled section to (A) without RAGE labeling with CD11b (red), vimentin (white), and DAPI for nuclei (blue). All images are taken in the distal alveolar region of the lung. Inset images boxed by white dotted lines are shown to the right of corresponding section. **(C)** Increased expression of immunophenotypic markers of cellular infiltrate in Vaped lung samples relative to Non-Vaped controls including CD45, CD206, CD11b, and CD11c. GAPDH is used as loading control. Corresponding quantitation of immunoblots is shown under each representative blot. n = 8 independent immunoblots using n = 4 non-vaped control mice and n = 17 vaped mice. Error bars represent SEM. ***$P < 0.001$, **$P < 0.01$. **(D)** Increased expression of immunomarkers of inflammation and matrix remodeling in Vaped lung samples relative to Non-Vaped controls including IL-6, IL-1B, high-mobility group box protein 1 (HMGB1), and fibronectin (FN). GAPDH is used as loading control. Corresponding quantitation of immunoblots is shown under each representative blot. n = 12 independent immunoblots using n = 4 non-vaped control mice and n = 17 vaped mice. Error bars represent SEM. ***$P < 0.001$.
Source data are available for this figure.

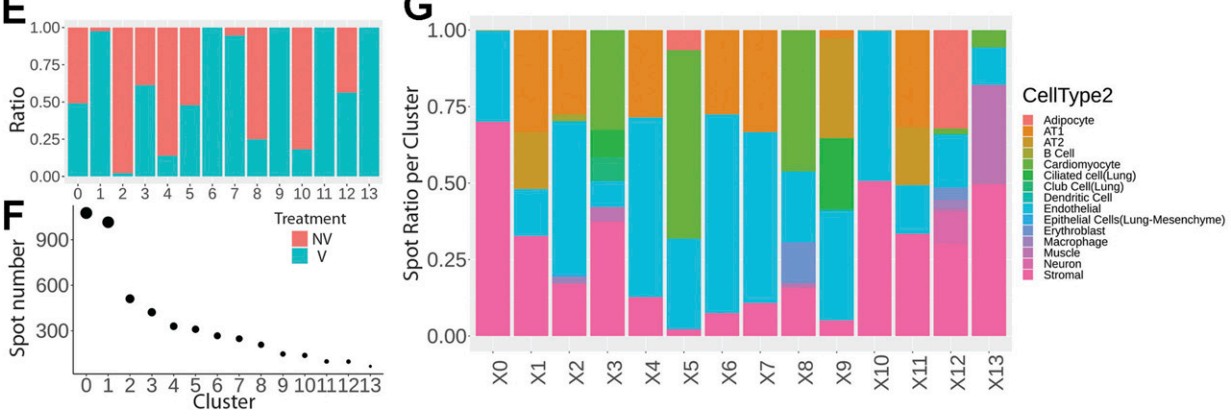

**Figure 6. Vaping induces spatial transcriptional changes in pulmonary tissue.**
**(A)** Schematic of experimental workflow representing capture of lung tissue within fiducial frame of gene expression slide, RNA library preparation and data analysis to spot annotation on the Mouse Cell Atlas database. **(B)** Spatial representation of unsupervised clusters overlaid on hematoxylin and eosin (H&E) micrographs of Non-Vaped and Vaped samples. **(C)** UMAP projection color-coded according to unsupervised clustering of gene signatures. **(D)** Identification of parenchyma and upper airway clusters overlaid on H&E micrographs of Non-Vaped and Vaped samples. **(E, F)** Relative and absolute (F) spot contributions of Non-Vaped and Vaped derived samples to each cluster as shown in UMAP (Panel 6C and Fig S12). **(G)** Putative cell type contributions by cluster relative to Mouse Cell Atlas database (Condense plot).

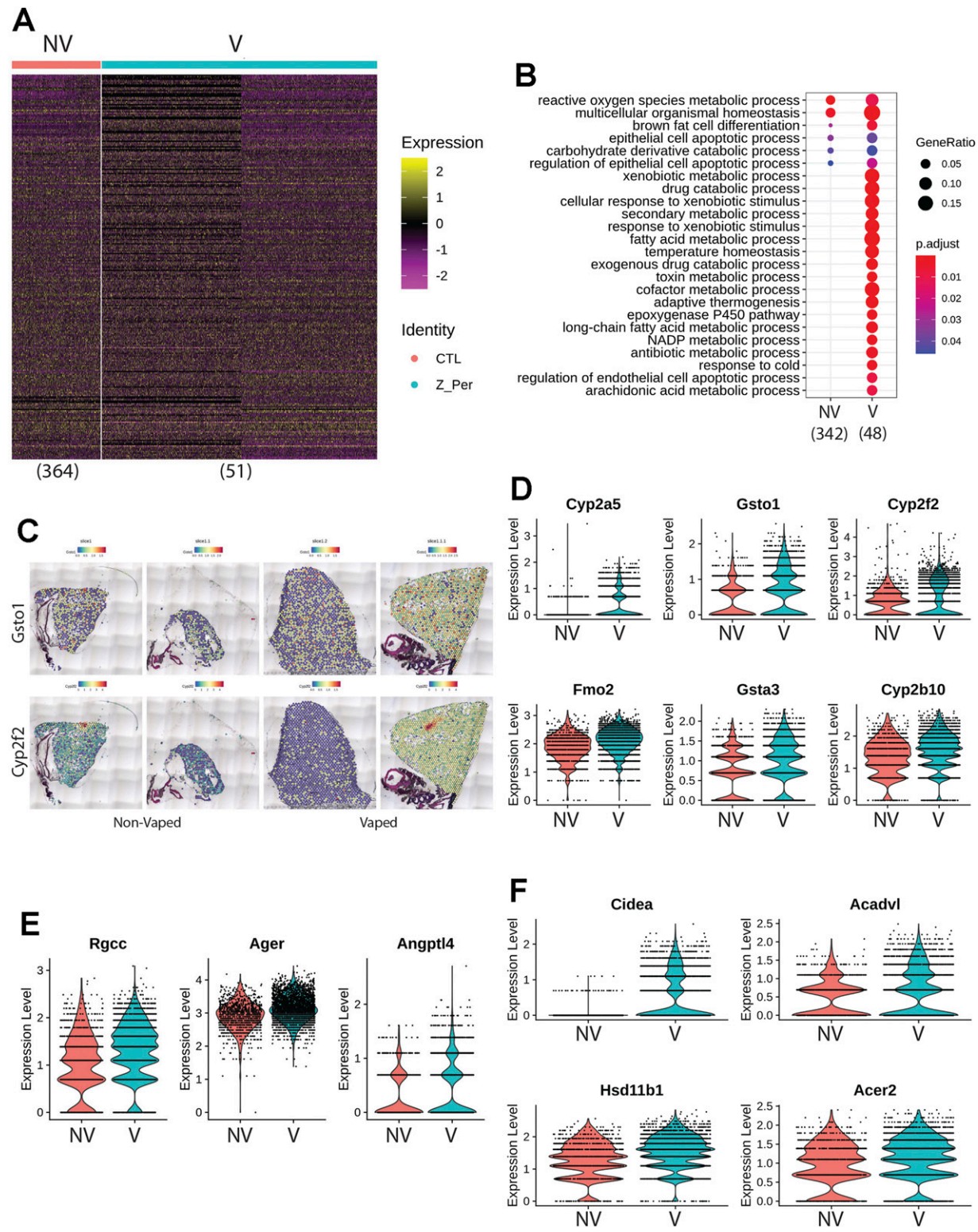

**Figure 7. Chronic vaping prompts response to xenobiotic stimulus, endothelial apoptosis, and lipid catabolism in lung parenchyma.**
**(A)** Heat map representing the differential expressed genes from Non-Vaped and Vaped samples in the parenchyma subset. **(B)** GO terms results from Gene Ontology analysis annotated by Biological Process. Circle diameter represents the gene ratio, whereas significance level is color-coded according to heat map scale. **(C)** Spatial expression and distribution of Gsto1 and Cyp2f2 in Non-Vaped and Vaped samples. **(D, E, F)** Violin plots indicating the single spot distribution and expression of gene targets of GO terms: (D) cellular response to xenobiotic stimulus, (E) endothelial cell apoptotic process, and (F) lipid catabolic process.

and Apoc1), and hydrogen peroxide metabolic process (GO: 0042743; Cyp1a1, Hbb-bt, and Prdx6) (Fig S8). Collectively, these results demonstrate parenchyma exhibit transcriptional changes associated activating processes of endothelial apoptosis and lipid catabolism in respond to vaping xenobiotic stimulus.

### Upper airway up-regulates transcriptome markers representing mitochondrial alterations during apoptosis and response to oxidative stress after vaping exposure

Differential expression analysis of the upper airway revealed 96 DEGs in the NV group and substantial up-regulation in the V group accounting for 1,385 DEGs (Fig 8A). GO analysis demonstrated enrichment of muscle contraction targets and processes in the NV group, with more metabolic GO terms in the vaped samples (Fig S9). Examination of biological processes revealed an enrichment of various ontologies associated with cellular stress on the vaped samples (Fig 8B). Biological significance of each GO term was assessed by comparing the gene count per GO term in addition to the gene ratio, verifying the DEG enrichment on the NV group (Fig 8C). Expression patterns of genes associated to the response to oxidative stress GO term (GO: 0006979) were consistently up-regulated in the upper airway of vaped samples (Fig 3D). GO terms associated to mitochondrial homeostasis, membrane organization, fusion, and fission and gene expression were pronouncedly higher in gene ratio and gene count per GO term in the V samples (Fig 8E and F). Consistent with the GO analysis, average expression of targets belonging to the apoptotic mitochondrial changes were higher in the upper airway of vaped samples (Fig 8G). Similar to results observed in the lung parenchyma, gene ratios and counts of GO terms associated to lipid metabolism were up in the V group, with up-regulated expression of genes in GO term (GO: 0016042) (Fig S9). Cross-referencing the DEGs from NV and V groups revealed an intersect of 80 DEGs characteristic of NV samples regardless of spatial distribution and 36 DEGs up-regulated in the V group (Fig S10). Together, these results demonstrate up-regulation of lipid catabolism and activation of apoptotic mitochondrial alterations and response to oxidative stress are the main transcriptional responses in the upper airway after vaping exposure.

### Pulmonary dysfunction and right ventricular remodeling in vaped mice

VAPI as evidenced by multiple independent criteria and analyses (Figs 1–8) demonstrates severe damage to lung tissue. The clinical syndrome of severe VAPI has been associated with cardiac dysfunction (15, 16, 50) prompting assessment of myocardial structure and function in vaped mice after the 9-wk termination point of exposure. Short axis echocardiography was performed on all mice before euthanasia (Fig 9A) and multiple parameters of cardiac structure and function were calculated. Significant worsening ($P <$ 0.001) of ejection fraction (EF; 62.41 ± 1.71 versus 74.91 ± 4.35), fractional shortening (FS; 32.67 ± 1.44 versus 42.45 ± 3.02), left ventricular interior diameter at systole (LVIDs; 2.129 ± 0.15 versus 1.794 ± 0.2), and left ventricular volume at systole (LV Vol S; 15.14 ± 2.25 versus 9.764 ± 3.67) was evident comparing hearts from vaped mice to non-vaped controls, respectively (Fig 9B–E). Interestingly,

these same four indices of EF, FS, LVIDs, and LV Vol S also exhibit a lesser but significant gender difference ($P <$ 0.05) subjects with consistently worse values for hearts of male versus female vaped mice (Fig 9B–E). All remaining calculated parameters of cardiac structure and function from echocardiographic measurements consisting of left ventricular mass, left ventricular volume at diastole, left ventricular interior diameter at diastole, intraventricular septum at diastole, intraventricular septum at systole, left ventricular posterior wall at diastole, and left ventricular posterior wall at systole were not significantly different (Table S2). Pulmonary circuit failure associated with VAPI cannot be readily assessed by echocardiography in mice, so coronal sections of hearts were evaluated to determine right ventricular structural abnormalities. Right ventricular chamber enlargement and wall thinning is consistently evident in hearts of vaped mice compared to nonvaped controls (Fig 9F and G). Furthermore, histologic quantitation of cardiomyocyte cross-sectional length reveals significant elongation of cells in the right ventricular wall of vaped mice compared with non-vaped controls ($P <$ 0.001 to 0.05; Fig 9H and Table S3). Pathological damage associated with right ventricular remodeling is also observed with increased collagen deposition at perivascular regions near the atrioventricular junction in hearts of vaped mice compared with non-vaped controls (Fig 9I and J). This preponderance of evidence indicating cardiac structural and functional deterioration pursuant to vaping is consistent with expectations of pulmonary failure in severe VAPI (15, 16, 50).

## Discussion

VAPI presents several novel and unprecedented challenges for researchers, clinicians, and society including (1) the diverse array of device delivery systems for vape aerosols including "do-it-yourself" modifications, (2) an incalculable number of formulations for vape juices also including "do-it-yourself" recipes, (3) variability in vaping profiles and topography among users, (4) inability to readily obtain target tissue samples and track pathogenic changes in vapers, (5) inherent biological differences between users including (but not limited to) age, gender, health status, and genetic composition, (6) limitations of experimental modeling systems to recapitulate biological conditions of human vaping, (7) rapidly evolving vape industry products, (8) ephemeral trends of the vape community, (9) inability to forecast long term effects of vaping due to lack of historical data (recency of development and utilization), and (10) balancing arguments of "harm reduction" for smokers versus creating a new generation of "never smoker" vapers. Assessing biological consequences of vaping in the human setting predominantly revolves around individuals experiencing symptoms of respiratory illness and relies upon non-invasive evaluation for diagnosis. Studying vaping pathogenesis requires understanding of alterations of both tissue structure/function as well as cellular/molecular biology that is not realistically possible in human subjects. As an alternative, rationally designed inhalation exposure models to provide biological insights are desperately needed. Paramount among the many questions to be addressed are: (1) what constituents of vape aerosol present substantial

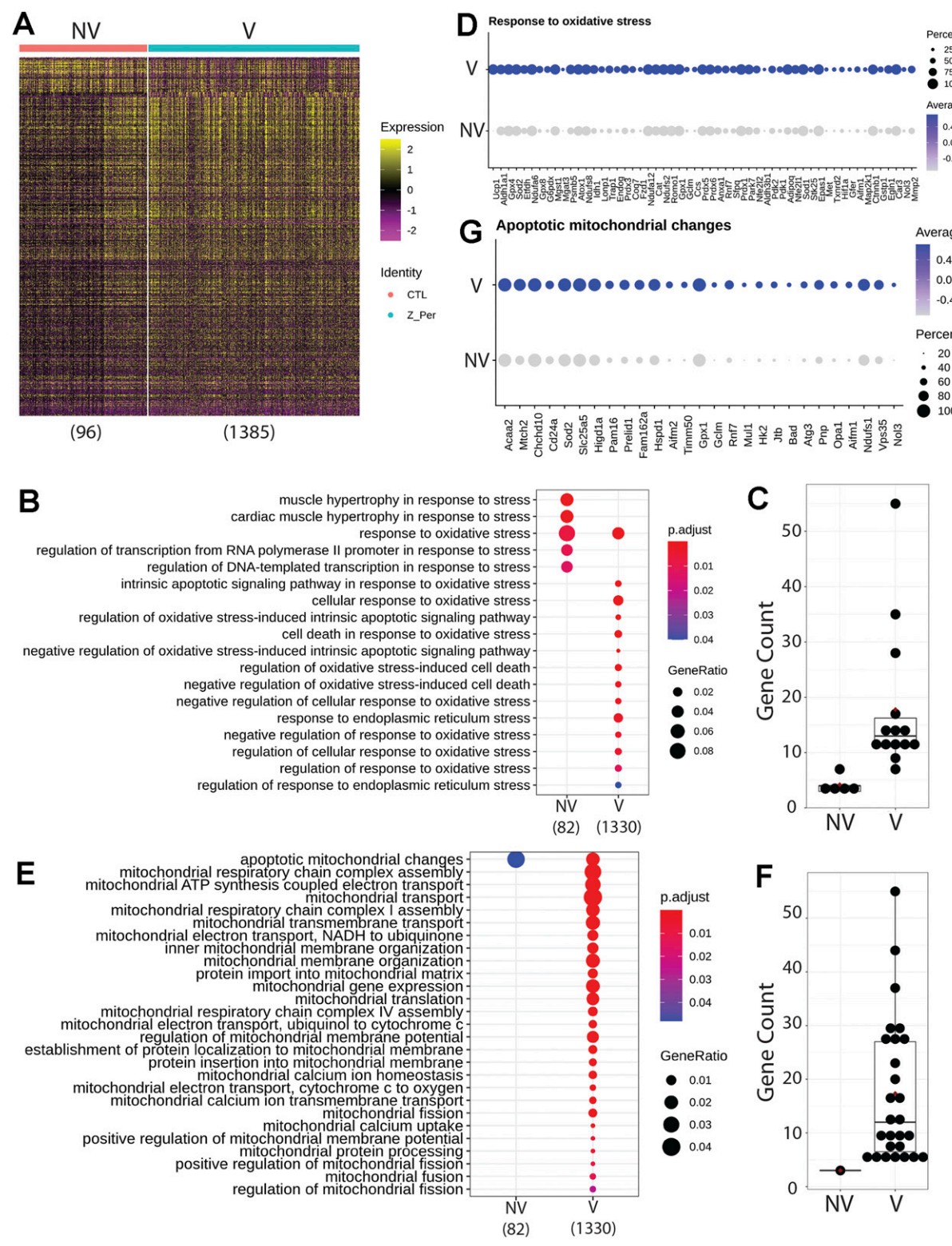

**Figure 8. Upper airway up-regulates targets linked to mitochondrial alterations during apoptosis and response to oxidative stress after vaping exposure.**
**(A)** Heat map representing the differential expressed genes from Non-Vaped and Vaped samples in the upper airway subset. **(B, E)** GO terms results from Gene Ontology analysis annotated by Biological Process and grouped by (B) cellular stress and (E) mitochondrial ontologies on Non-Vaped and Vaped samples. Circle diameter represents the gene ratio, whereas significance level is color-coded according to heat map scale. **(C, F)** Gene counts per GO term grouped by (C) cellular stress and (F) mitochondrial ontologies on Non-Vaped and Vaped samples. **(D, G)** Dotplot representing expression of marker of GO terms: (D) Response to oxidative stress and (G) Apoptotic mitochondrial changes. Circle diameter represents the percentage of spots expressing a particular gene, whereas normalized average expression is represented by color intensity.

hazards for triggering pathological processes, and (2) what are predisposing factors for severe VAPI manifestation in some individuals? Identifying characteristics of "at risk" users will be critical to avoid potentially life changing pulmonary damage, and biological assessments using patient samples could offer diagnostic value for clinically relevant injury status evaluation. Whereas most vapers do not experience overt symptoms of VAPI, let alone exhibit pulmonary failure, a small subset of users succumb to this type of injury (10, 11, 12, 13, 14, 15, 16, 17). Such severe VAPI cases arise in users of commercially available vaping products, unlike the EVALI epidemic in 2019 traced back to black market "Dank Vapes" associated with ill-conceived mixing of Vitamin E acetate and THC into vape juice (48, 49). EVALI syndrome often presented as acute respiratory distress syndrome with fulminant inflammation in the respiratory tract (4, 5, 6). In comparison, pathological processes culminating in severe VAPI are more obscure and slow to develop, in some cases occurring over months of frequent unrelenting vaping activity (10, 11, 12, 13, 14, 15, 16, 17). Thus, rather than model lung injury using xenobiotic agents that induce rapid and overwhelming inflammatory activity within hours such as bleomycin or meconium (68, 69), the conceptual framework for this study was to use widely available retail vaping technology and product combined with an extended exposure protocol consistent with topography of a high intensity vaper likely to develop VAPI leading to hospitalization.

Pathological outcomes of vaping vary dramatically in humans as well as laboratory models, but occurrence of severe VAPI in the clinical setting is a serious and growing concern in society (10, 11, 12, 13, 14, 15, 16, 17). The VAPI model developed in this report provides an experimental platform for this understudied syndrome, exhibiting a collective set of distinguishing characteristics without precedent in the literature. Foremost, phenotypic manifestations are reminiscent of clinical case reports of individuals hospitalized for severe VAPI including tissue damage, inflammatory responses, and cardiopulmonary failure. Importantly, pathogenesis was mediated by peach flavored vape juice with menthol cooling agent popularly used in the community delivered by the ubiquitous JUUL vape pen. Inhalation exposure of popular vape products using established vaping devices remains the best approximation of human activity to study biological effects upon the cardiopulmonary system. Vaping topography implemented in this report was deliberately designed based upon publications documenting human behavior and quantitative inhalation measurements (Fig S11; (39, 40, 41)). Puff frequency, duration, and flow rate were all set at values consistent with typified heavy vaping consumption for human subjects to the extent that our approach using an exposure chamber system can deliver similar parameters. Vaping topography is a critical factor in experimental design and varies widely between individual reports, often without detailed explanations for chosen parameters (35). Justification and/or standardization of vaping topography remains an important unresolved issue in most vaping-related research studies. Judicious implementation of puff topography for inhalation exposure using real-world products common in the vaping community produced a preclinical VAPI model over a 9-wk time course with clear evidence of pathological damage. Development of this inhalation exposure model enabled profiling of vaping biological effects on an unprecedented level using a combination of microscopic, biochemical, and molecular analyses offering fundamental insights into pathological responses in the cardiopulmonary system.

Pathological alterations consistent with VAPI in our murine model include multiple abnormalities in the respiratory tract. Widespread remodeling is evident from the upper airway passages to distal alveolar spaces deep within the lung parenchyma. Alveolar rarefaction, particularly evident in peripheral regions of the lung, is significant as demonstrated by mean linear intercept analysis (Fig 1). Deterioration of alveolar integrity is a hallmark of multiple respiratory distress syndromes including VAPI (70). Despite substantial loss of alveolar integrity the mice in this study did not exhibit overt symptoms of respiratory distress throughout the time course of inhalation exposure and there was no occurrence of adverse events. Additional physiological analyses of respiratory functional parameters such as plethysmography and blood oxygenation level (pO2) could reveal subclinical manifestations in mice not apparent from behavioral observation. Thickening of bronchiolar passages together with localized accumulation of cellular infiltrate are histologic findings consistent with chronic irritation (Fig 1). Degeneration of airway structural integrity including features of epithelial dysmorphia as well as disruption of extracellular matrix were readily observed in histological evaluation (Fig 2). Elastin disorganization and increased collagen deposition presumably decrease airway compliance as has been described in other respiratory injury syndromes (71, 72). Elevation of mucin production and secretion represents a defensive mechanism within airways to facilitate clearance of inhaled foreign agents (55, 56) and is clearly a feature of our VAPI model (Fig 3). Vape juice exposure provokes various biological responses including increased secretory activity (56), but relative roles of the VG/PG vehicle versus minor component additives of flavorings and menthol cooling agents remains obscure and continues to be a focus of ongoing investigation (24, 25, 47, 73, 74). Induction of inflammation and elevation of cytokine production (Fig 5) is commonly present in lung injury (25, 75, 76, 77), but cellular infiltrates in our VAPI model are present in focal areas rather than uniformly distributed throughout tissues, most prominently proximal to airways or vessels indicative of remodeling. Chronic irritation serves as an inciting stimulus for oncogenic transformation (78, 79), and indeed sporadic histologic signs of early stage tumorigenesis were rare events but present in our VAPI model (data not shown). Among lung epithelial proteins, ECAD plays particularly important roles in maintenance of cellular junctional integrity and has been linked to immunomodulatory activity (59). Thus, the profound ECAD up-regulation in our VAPI model (Fig 4) may be multipurpose and it is tempting to speculate that ongoing suppression of airway inflammation may be involved. The potential relationship between VAPI, chronic airway inflammation, immunomodulation by ECAD, and oncogenic risk is an intriguing possibility to be explored in future studies.

Transcriptomic profiling is a powerful tool for identifying cellular reprogramming consequential to environmental stress. Transcriptional profiling of vape fluid exposure has primarily involved cell culture models examining acute changes after short-term exposure protocols (80, 81, 82, 83, 84). Such studies have implicated vape juice for induction of genes associated with inflammation, metabolic/biosynthetic processes, extracellular membrane, apoptosis, lipid

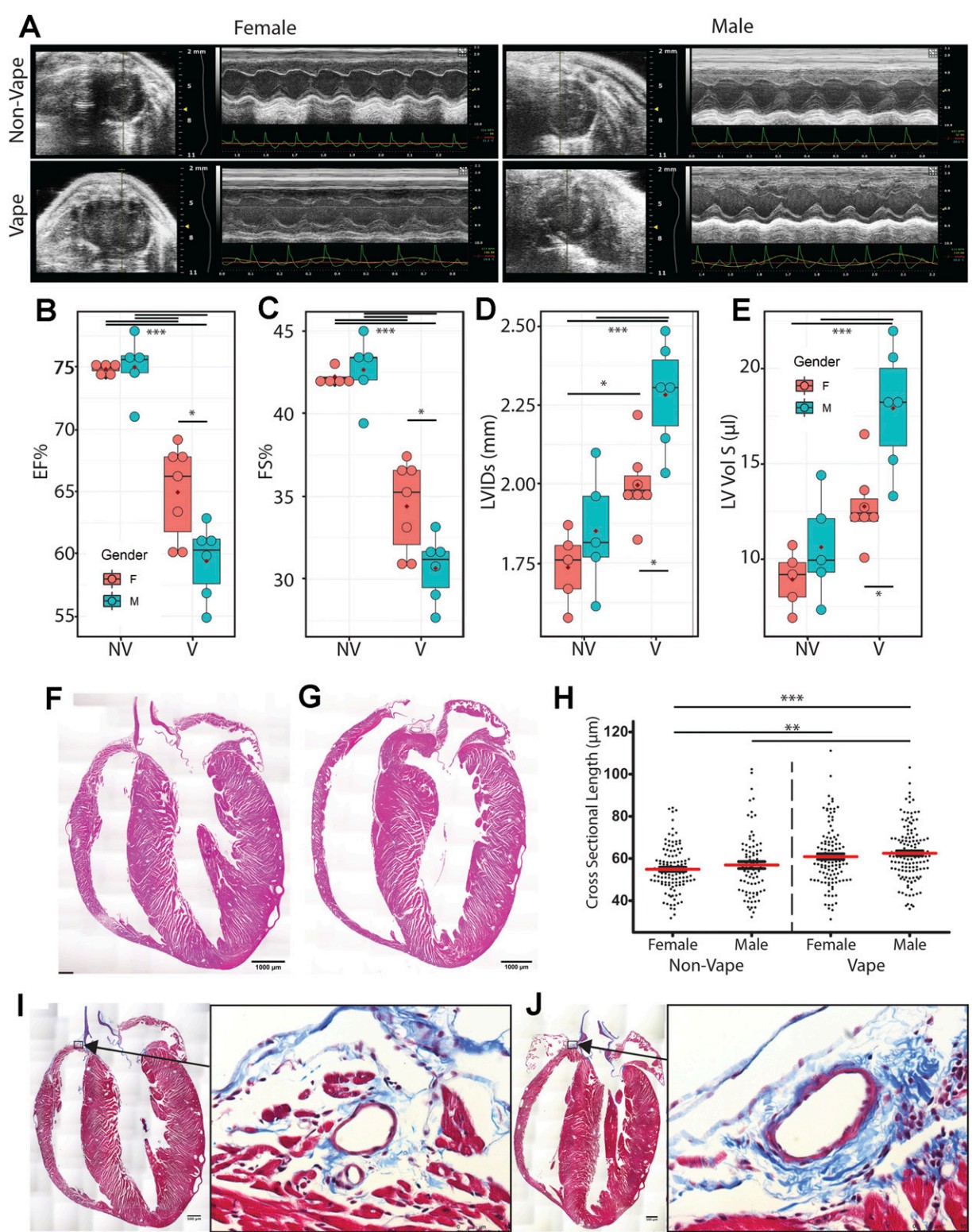

**Figure 9. Pulmonary dysfunction and right ventricular remodeling in vaped mice.**
**(A)** Representative 2D echocardiography images (M-mode) at study completion (week 9). Para-sternal short-axis view showing LV anterior wall and posterior wall movement. **(B, C, D, E)** Echocardiogram data from non-vaped and vaped samples measuring (B) ejection fraction, (C) fractional shortening, (D) left ventricular interior diameter in systole and (E) left ventricular volume in systole. **(F, G)** Hematoxylin and eosin (H&E) micrographs of (F) Non-Vaped and (G) Vaped hearts in coronal view. **(H)** Myocyte cross-sectional length in right ventricle. **(I, J)** Masson's Trichrome stain of (I) Non-Vaped and (J) Vaped samples hearts in sagittal view (Arrow: collagen deposition. Arrowhead: cellular infiltrate). ANOVA with Kruskal–Wallis significant differences test. $P < 0.05$(*), $P < 0.01$(**), $P < 0.001$(***).

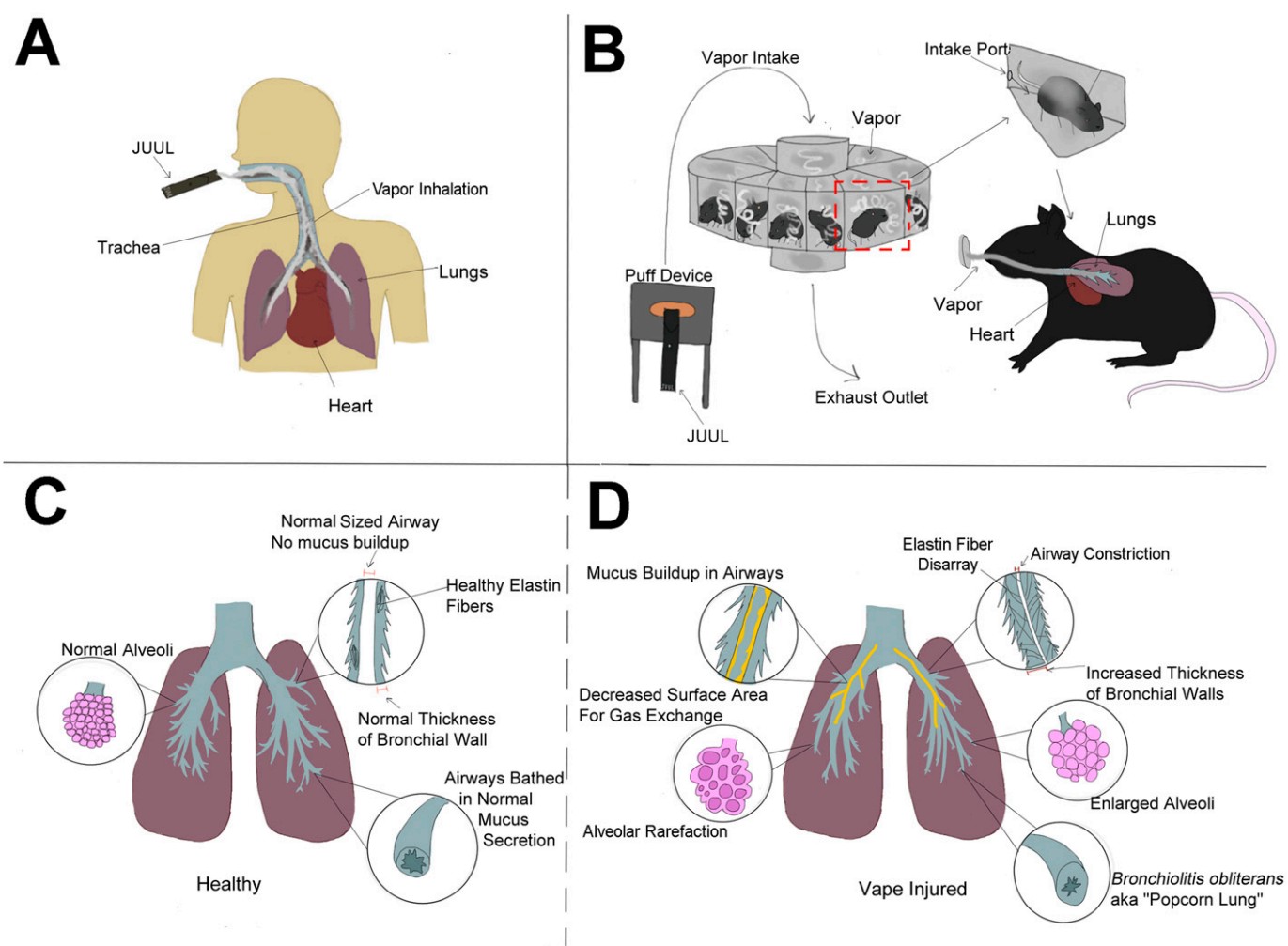

**Figure 10. Vaping-associated pulmonary injury modeling and pathology.**
**(A, B, C, D)** Schematic representation comparative summary of inhalation exposure between humans (A) versus the InExpose murine model (B), and healthy lung biology (C) versus pathological changes observed in this study (D). Additional pathological changes in right ventricular structure leading to pulmonary failure were also observed, but are not represented in the diagram.

metabolism, and hypoxia. None of these studies evaluated an inhalation exposure model of VAPI, which presents challenges for transcriptome profiling. Respiratory tract cellular composition is profoundly distinct in the upper airway compared with lung parenchyma, so spatial transcriptomics was used to facilitate regional identification of profiles. Predominant effects at the transcriptional level in lung parenchyma include processes associated with endothelial apoptosis and lipid catabolism in respond to the xenobiotic stress of vape fluid exposure. In comparison, transcriptomic changes in the upper airway involve up-regulation of lipid catabolism and activation of apoptotic mitochondrial alterations and response to oxidative stress in the vaped samples relative to non-vaped controls. Select DEGs identified in bioinformatic analyses of particular interest include Gsto1 and Cyp2f2 because of their roles in inflammation (85), club cell differentiation, lung epithelial regeneration (86), and metabolism and toxicity of xenobiotic compounds (87). Rgcc, Ager, and Angptl4 were among the up-regulated DEGs within the endothelial cell apoptotic process GO term (GO: 0072577). Ager, the gene encoding for RAGE is a key marker in pathophysiology of chronic obstructive pulmonary disease associated with cigarette smoking (88, 89, 90) (Fig 7E). These few examples demonstrate the diverse transcriptional effects characteristic of VAPI in our murine model. Broad roles of gene families mediating responses to xenobiotic agents, reactive oxygen species, lipid metabolism, and apoptotic signaling identified in this report provide a roadmap for future characterization in human VAPI samples. The differential expression analysis strategy of Model-based Analysis of Single-cell Transcriptomics (MAST) test (91) did not yield DEGs in our analysis. However, applicability of the MAST test to spatial transcriptomic data relative to the features and assumptions used for single cell data are unclear and possibly inappropriate. We suspect the multimodality and sparsity of spatial transcriptomic data from highly admixed tissues as the lung, which holds the transcriptome of numerous mixed cells within a 50-$\mu$m spot does not compare to single-cell transcriptome derived from a microfluidics approach. Nevertheless, DEG results are valid based upon expression levels per spot and spots count of targets of interest that were robust and consistent with Wilcoxon test results (Fig 7D, 8D and G, S8, and S9). Candidate

genes could be valuable for screening patient samples to serve as diagnostics for assessment of VAPI severity in clinical setting, perhaps in subclinical evaluation of patient risk for VAPI. The extraordinary sensitivity and wealth of information derived from spatial transcriptomics allowed for an unprecedented view of cellular reprogramming in response to vape fluid inhalation exposure with the resulting datasets and their bioinformatic analyses forming the roadmap for future investigation.

Severe VAPI involves pulmonary failure including cardiac structural remodeling and functional decline (15, 16, 50). Echocardiographic assessment provided initial indications of myocardial involvement in VAPI based upon significant decreases in EF and FS as well as increases in LVIDs and LV Vol S (Fig 9). Left ventricular functional and structural alterations are consistent with modest systemic hemodynamic decline rather than heart failure, as would be expected for VAPI with primary impact upon pulmonary circulation. Because assessment of right ventricular structure is not feasible by echocardiography in mice, evaluations were performed by histological analyses of coronal sections as well as measurement of cardiomyocyte length. Pathologic alterations of right ventricular structure include chamber enlargement and wall thinning as well as increased perivascular collagen deposition (Fig 9J), all indicative of increased hemodynamic stress in the pulmonary circuit. Increased collagen deposition surrounding the vessel above the right ventricle (Fig 9J) is consistent with increased perivascular collagen in lung sections (Fig 2). Right side failure is likely due to persistent pulmonary hypertension, but measurement of murine right ventricular hemodynamics is not practicable. Implications of pulmonary failure including impaired pO2 as well as compromised exercise capacity resulting from VAPI are the subject of ongoing studies. Increased cardiomyocyte length in the right ventricular free wall of mice suffering from VAPI compared with non-vaped controls is consistent with chamber dilation and wall thinning (92, 93). Collectively, cardiac remodeling is consistent with expectations for severe VAPI-pathogenesis with one unanticipated distinction: the consistent gender-associated difference in severity. Female cardioprotection from pathological challenge is mediated by estrogenic activity involving AKT/Pim-1 axis downstream signaling (94, 95, 96, 97). Ramifications of gender-based differences in cardioprotection in relation to risk of pulmonary involvement will be an interesting biological phenomenon to correlate with clinical observations as VAPI hospitalizations accrue in the future.

No model can completely capture the complexities of human vaping exposure, but our murine model of inhalation exposure using popular vaping products and puff protocols based upon studies of human experience brings a new platform that can be modified as desired for rational experimental design (Fig 10A–D). Moreover, findings in this report provide broad-based yet integrated depiction of VAPI at the tissue, cellular, and molecular level serving as a reference point for future research. There is a desperate need for rigorous, unbiased, and independent research studies assessing biological responses to vape aerosol with inhalation exposure experimental models. Society is perched on the precipice of a new era for ENDS devices and vaporizers. Big Tobacco companies continue to find innovative ways to market tobacco-related devices and products. Explosive growth of vaping over the past decade demonstrates consumer willingness to adopt new technologies for inhalation exposure in the absence of rigorous scientific research on potential health risks. Heated Tobacco Products represent the next generation of devices from Big Tobacco companies being touted for "harm reduction" compared to combustible cigarette smoking. Identical arguments of decreased health risks continue to be made for ENDS devices, and yet the evidence of pathological processes consequential to vaping is accumulating from past and ongoing research as well as clinical findings. Admittedly, there will always be a spectrum of VAPI from subclinical to mild to severe. However, no one can predict prevalence or clinical presentation of VAPI in the years ahead with increasing years of exposure and advancing age of the user population. Only the passage of time will provide the information needed to assess the long term consequences of VAPI and recovery potential for human vapers, but these are certainly areas worthy of further investigation (98). Findings in this report are deliberately intended to reflect severe VAPI that represents a minority of vaping-associated respiratory illness. Modeling severe VAPI consistent with clinical reports (10, 11, 12, 13, 14, 15, 16, 17) provides insight regarding risk factors, biological responses, and cardiopulmonary susceptibility in a dynamic marketplace attracting a diverse user base that will continue to expand for the foreseeable future. Whether vaping technology or avant-garde heated tobacco products represent a lower risk for cardiopulmonary complications, respiratory illness, and cardiovascular diseases remains to be seen.

# Materials and Methods

### Mouse vaping inhalation protocol

4-wk-old male and female C57BL/6J mice were purchased from (Cat. no. 00064; Jackson Laboratory) and housed four mice per static cage. Ambient temperature was 70–72°F, on a 12-h light–dark cycle with automatic light control. Mice were supplied with Rodent Maintenance Diet (Teklad Global 14% Protein) and water ad libitum. During the adult phase of mouse life, 2.6 d are approximately equivalent to one human year. Vaping of mice started at ~8 wk (56 d) in a time course of 8–9 wk (age at conclusion equals 112–119 d). This represents the equivalent of human vaping from 16 to 37.5–40.2 yr of age according to Dutta and Sengupta (99) or the equivalent of 18–25 yr of age according to Flurkey (100). 6-to 8-wk-old mice were exposed to Peach Ice 70VG/30PG (ORGNX) flavored vape juice delivered as e-vapor from JUUL pens in whole body exposure chambers (inExpose; SCIREQ). Vaping chamber setup is represented schematically in Fig S11. Mice were exposed to 3 s puffs every 20 s at 1.8 liters/minute, intake rate. Exhaust pumps for fresh air flow rate was 2.5 and 1.5 liters/minute for the 4-h duration of the vaping profile (Fig S11) based upon human vaping topography recommended by Farsalinos et al. concluding "4-s puffs with 20–30 s interpuff interval should be used when assessing electronic cigarette effects in laboratory experiments, provided that the equipment used does not get overheated." (41) In addition, puff duration parameters and frequency are within the reported range of human vaping topography from real time characterization of electronic cigarette use in the 1 Million Puffs Study (40). Animals were exposed for 4 h/day, 5 d/week for 9 wk (Fig S11).

### Histological staining

Lungs of e-cigarette-exposed and control mice were harvested by inflation with formalin. Right lung lobes were sutured off at the right primary bronchus and frozen back in liquid nitrogen immediately after dissection, whereas the left lobe was manually inflated over the course of 15 min with ~1.5 ml of formalin until visual confirmation of sufficient inflation. The left lobe was then dissected and submerged in formalin for 24 h followed by tissue processing, paraffin embedding, and sectioning. The lungs were compressed to attain the primary bronchial tree in the same plane of view as parenchyma. Sections were stained with Harris Hematoxylin and Eosin-Phloxine (H&E) in addition to Movat's Pentachrome reagents to visualize morphometric and structural changes. Lung sections were stained after the modified Russell-Movat pentachrome stain protocol to visualize changes in collagen, elastic fibers, and mucin deposition. Pentachrome staining is interpreted as elastic fibers (black to blue/black), nuclei (blue/black), collagen (yellow to red), reticular fibers (yellow), mucin (bright blue), fibrin (bright red), and muscle (red). Trichrome Stain (Masson) Kit (Cat. no. HT15; Sigma-Aldrich) was used to visualize collagen deposition in hearts and lungs according to the manufacturer's protocol. Images were acquired using a Leica DMIL6000 microscope running XY stage tile scanning and subsequently stitched using ImageJ software. PAS stain: Slides were deparaffinized and rehydrated after generic procedures. PAS stain was performed following the kit protocol specifications (1.01646.0001; Sigma-Aldrich). Hematoxylin solution modified according to Gill III was substituted, as recommended by the protocol, for hematoxylin solution modified according to Gill II (GHS216; Sigma-Aldrich). Finally, slides were mounted with toluene solution (Lot#103929; Thermo Fisher Scientific). Images were acquired using a Leica DMIL6000 microscope. All images were taken in the upper airway as close as possible to the branch point of the primary and secondary bronchi.

### Quantitative analysis of lung morphometry by mean linear intercept and bronchial wall thickness

#### *Mean linear intercept quantitation*
H&E–stained sections of e-cigarette exposed and control mice were analyzed for changes to the mean free distance between gas exchange surfaces, denoted as mean linear intercept (Lm; N = 10 for each group). 10 randomly selected regions of lung parenchyma without bronchioles or vessels from each mouse were imaged and analyzed using the semi-automated method developed by Crowley et al ([101]). 10 test lines were superimposed over the images and chords between alveolar walls were measured. An average of 2,000 chords per mouse were obtained. Lm was calculated by multiplying the lengths of the chords by the number of chords and dividing the product by the sum of all the intercepts.

#### *Bronchial wall thickness*
Lung sections of e-cigarette exposed and control mice stained with H&E were analyzed for changes to the thickness of the bronchial walls. Images of the bronchioles in a cross-sectional plane of view were taken from each mouse and measured using ImageJ software. An average of six independent measurements per mouse were taken. The total area of the bronchial was measured by tracing the outside of the basement layer of the epithelium and the area of the lumen was measured by tracing the inner border of the epithelium. Wall area percentage was calculated by dividing the difference of the total area and the lumen area by the total area.

### Immunohistochemistry and confocal microscopy

Tissue samples were formalin fixed and paraffin embedded as previously described (see "Histological staining") ([102]). Paraffin sections (5 $\mu$m) were deparaffinized, subjected to antigen retrieval in 10 mM citrate, pH 6.0, and quenched for endogenous auto-fluorescent activity in 3% sodium borohydride (452882-250; Sigma-Aldrich) in TN buffer (150 mM NaCl and 100 mM Tris, pH 7.6) for 30 min. Blocking was performed for 30 min at room temperature with 10% Horse Serum in 1xTN. Tissues were immunolabeled with primary antibodies as listed in Table S1 overnight at 4°C. Fluorescently conjugated secondary antibodies were diluted 1:200 (Donkey anti-goat 488 Cat. no. A11055; Life Tech, Donkey anti-rabbit 555 Cat. no. A32794; Invitrogen, Donkey anti-rat 555 Cat. no. 712-166-153; Jackson Labs, Donkey anti-chicken 555 Cat. no. 703-165-155; Jackson Labs, Donkey anti-rabbit 647 Cat. no. A31573; Invitrogen, Donkey anti-chicken 647 Cat. no. 703-505-155; Jackson Labs). DAPI (Cat. no. 62248; Thermo Fisher Scientific) was applied in the final wash step at 0.1 $\mu$g/ml to label nuclei. Images were acquired using a Leica SP8 confocal microscope and processed with Leica and Photoshop software. Stains were performed on at least three different samples per exposure group, and one technical replicate. Tissue was imaged from trachea, conducting airways, and lung parenchyma.

#### *Muc5ac immunostain*
Frozen sections prepared identical to those used for spatial transcriptomics (see below) were used to visualize Muc5ac. Slides stored at –80°C were warmed to room temperature and washed twice in PBS (0.37 M NaCl, 27 mM KCl, 100 mM Na$_2$HPO$_4$, and 18 mM KH$_2$PO$_4$), then fixed in –20°C methanol for 30 min. Two more washes in PBS were performed, and then the tissue was blocked using 10% Horse Serum for 30 min. Next, two washes in PBS were followed by immunolabeling with Muc5ac (Cat. no. 1364248; U.S. Biotech) and ECAD (Cat. no. AF748; R&D Systems) overnight at 4°C. Two more washes in PBS were performed the following day, and fluorescently conjugated secondary antibodies were used to detect primary antibodies (Donkey anti-Rabbit 555 703-165-155; Jackson Labs, Donkey anti-Goat A11055; Life Tech) for 1 h and 30 min at room temperature. After two washes in PBS, DAPI (Cat. no. 62248; Thermo Fisher Scientific) was applied in the final wash step at 0.1 $\mu$g/ml to label nuclei for 15 min in PBS, and slides were coverslipped and mounted in Vectashield (Cat. no. H-1000; Vector Labs). Images were acquired using Leica SP8 confocal microscope and images were processed with Leica software. Tissue was imaged in most distal airways where mucin secretion is uncommon.

### Immunoblotting

Lungs from vaped animals and air-exposed controls were collected and stored at –80°C for future analysis. Frozen lung tissues were homogenized in lysis buffer containing protease and phosphatase

inhibitor cocktails (P8340, P5726, and P2850; Sigma-Aldrich). Bradford assay was performed to analyze and normalize protein concentrations and lysates were prepared by addition of NuPAGE LDS Sample Buffer (NP0007; Thermo Fisher Scientific) and 100 $\mu$M dithiothreitol (161-0611; Bio-Rad). Samples were sonicated and boiled for 5 min at 95°C and stored at –80°C. Proteins were separated on a 4–12% NuPAGE Bis-Tris gel (NP0321BOX; Thermo Fisher Scientific) and transferred onto a polyvinylidene fluoride membrane. Nonspecific binding sites were blocked using Odyssey blocking buffer (927-60001; LICOR) and proteins were labeled with primary antibodies in 0.2% Tween in Odyssey blocking buffer overnight. After multiple washes, blots were incubated with secondary antibodies in 0.2% Tween 20 in Odyssey blocking buffer for 1 h 30 mins at room temperature and scanned using the LICOR Odyssey CLx scanner. Quantification was performed using ImageJ software. Antibodies and their dilutions are listed in Table S4.

## Spatial transcriptomic analyses

### Sample preparation
Lungs were inflated through the trachea with a 1:1 PBS/OCT mix (1× PBS, Cat. no. 21-040-CV; Corning; TissueTek O.C.T. Compound, Cat. no. 25608-930; VWR) with RNAse inhibitor at 0.2 U/$\mu$l (Cat. no. 3335399001; Millipore Sigma). Tissues were embedded in OCT and flash frozen using an isopentane and liquid nitrogen bath. Tissue sections of 10 $\mu$m were obtained on a CryoStar NX70 (Thermo Fisher Scientific) and processed immediately for spatial transcriptional analysis or stored for histological stains. Blocks and sections were maintained at –80°C for long-term storage.

### Image collection and spatial transcriptomic library preparation
Freshly obtained cryosections were placed in Visium gene expression slides (Cat. no. 2000233; 10X Genomics) for processing. Tissue staining with hematoxylin and eosin and image collections were performed as recommended by the Visium protocol. Images were collected on a Leica DMI6000 B on a 5× objective at a 1.16 $\mu$m/pixel capture resolution (Fig S12). Spatial transcriptomic libraries were prepared using Visium Spatial Gene Expression Slide & Reagent Kit following the manufacturer's protocol (PN-1000184; 10X Genomics). Lung permeabilization time was optimized Visium Spatial Tissue Optimization Slide & Reagent Kit (PN-1000193; 10X Genomics). Samples were processed together to avoid introduction of technical batch effects. Library concentration and fragment size distribution of each library were tested with Bioanalyzer (Agilent High Sensitivity DNA Kit, Cat. no. 5067-4626; average library size: 500–610 bp). The sequencing libraries were quantified by quantitative PCR (KAPA Biosystems Library Quantification Kit for Illumina platforms P/N KK4824) and Qubit 3.0 with dsDNA HS Assay Kit (Thermo Fisher Scientific). Sequencing libraries were submitted to the UCSD IGM Genomics Core for sequencing (NovaSeq 6000), aiming for >50K reads per spot.

### Spot processing and quality control
Raw data were processed on the SpaceRanger pipeline (10X Genomics; version 1.2.2, Figs S13–S16). Sequencing reads were aligned to the 10x mouse genome mm10-2020-A. Spots maintained comparable UMI and gene count detection throughout the tissue section (Fig S2). Counts per spot were normalized to account for variance within tissue anatomy, and transformed using Seurat R Package (v4.0.3) to account for technical artifacts and preserve biological variance (103, 104). Preparations derived from slide preparation yielded 4933 barcoded spots for analysis, from which 1,315 corresponded to Control (842 male, 473 female), and 3,618 corresponded to 0% vape group (1,565 male, 2,053 female). Final removal of unwanted sources of variation and batch effect corrections was performed using Seurat R Package (v4.0.3).

## Bioinformatics

### Dimensionality reduction and unsupervised clustering
The first 30 principal components (PCs) were used to perform dimensionality reduction. Approximately 1,000 spatially variable genes were selected based on their expression and dispersion using the "markvariogram" method (105). PC analysis was performed on the scaled data as a dimensionality reduction approach. The first 30 PCs were selected for unsupervised clustering and non-linear dimensional reduction (UMAP; Fig S3). Clusters were classified using the clustifyr package (106) using the Tabula muris data as reference set for classification (107) (data not shown), and spots mapped to the MCA database using scMCA package (108).

### GO analysis
GO enrichment analysis for DEGs lists derived from parenchyma and upper airway spots was performed using the enrichGO and compareCluster functions of clusterProfiler (3.16.1) R package (109). Bonferroni correction was use as a multiple hypothesis test method to control the number of false positives (110).

## Echocardiography

Transthoracic echocardiography was performed on lightly anesthetized mice under isoflurane (1.0–2.0%; Abbot Laboratories) using a Vevo 2100 (VisualSonics). Hearts were imaged in the 2D parasternal short-axis (SAX) view, and M-mode echocardiography of the mid-ventricle was recorded at the level of papillary muscles to calculate FS. From the recorded M-mode images, the following parameters were measured: EF, left ventricular (LV), anterior wall thickness (AWT), LV posterior wall thickness (PWT), LV internal diameter (LVID), and LV volume in diastole (index: d) and systole (index: s).

## Cardiac histology

After anesthetization of the mice by ketamine, hearts were arrested in diastole and perfused with formalin for 15 min at 80–100 mm Hg via retrograde cannulation of abdominal aorta. Retroperfused hearts were removed from the thoracic cavity and fixed overnight in formalin at room temperature. The hearts were processed for paraffin embedding and sectioned in the sagittal orientation at 5 $\mu$m thickness at room temperature. The heart sections were stained with Harris hematoxylin and eosin Phloxine to visualize morphometric and structural changes. Images were obtained by a

Leica DMIL6000 microscope using XY stage tile scan and manually stitched using ImageJ software.

### Cardiomyocyte cross sectional area quantitation

Heart sections of all treatment groups were acquired and stained as previously described with the exception that heart samples were not treated with sodium borohydride. 24 images of each right ventricle were taken using a Leica SP8 confocal microscope at a 400× magnification. Quantification of the cross sectional area of right ventricular myocytes was done in ImageJ software using a pixel/um ratio of 3.5 for all images analyzed. Myocytes that contained two clear intercalated discs at each end with an associated nucleus were measured by drawing a line from one intercalated disc to the other, then measuring the length of the line. All measurements taken from each experimental group are listed in the table below. Data points were input into Prism5 software to compose a graph and run a one way ANOVA test using Kruskal–Wallis metrics, $P < 0.01$. Male Vape (n = 4, 133 measurements, SD: ±12.9 $\mu$m). Female Vape (n = 4, 116 measurements, SD: ±14.13). Male No Vape (n = 3, 83 measurements, SD: ±14.97). Female No Vape (n = 3, 72 measurements, SD: ±9.935).

### Statistics

For mean linear intercept and bronchiole wall area percentage, unpaired $t$ tests were performed between the vape and no vape groups with a 95% confidence interval and a two-tailed $P$-value of 0.0167 for 1D and 0.0098 for 1E I using GraphPad Prism Version 5.02. For echocardiography, 5–7 mice per group ANOVA with Kruskal–Wallis significant differences test with $P < 0.05$(*), $P < 0.01$(**), $P < 0.001$(***). For differential expression analysis, Wilcoxon rank sum test was performed with selection for a threshold of 0.05 for an adjusted $P$-value and a log (FC) > 0.25 was used to define statistically significant and DEGs.G0 term analysis was performed with $P$-value cutoff of 0.05 using Benjamini–Hochberg Procedure. For immunoblot analysis, two-tailed unpaired $t$ test was used to compare two groups of vape and non-vape samples. Statistical analysis was performed using GraphPad Prism. A $P$-value of < 0.05 was considered statistically significant. For cardiomyocyte cross-sectional length, right ventricular myocytes were analyzed in male and female, vaped and non-vaped samples. A one-way ANOVA using Kruskal–Wallis $t$ test metrics was performed on the total number of measurements per group ($P < 0.01$). Male Vape (n = 4, 133 measurements, SD: ±12.9 $\mu$m). Female Vape (n = 4, 116 measurements, SD: ±14.13). Male No Vape (n = 3, 83 measurements, SD: ±14.97). Female No Vape (n = 3, 72 measurements, SD: ±9.935).

### Study approval

Animal protocols and experimental procedures were approved by the Institutional Animal Care and Use Committee at San Diego State University.

## Data Availability

Spatial transcriptomic data generated in this study has been uploaded to the Gene Expression Omnibus database (GSE188805).

## Supplementary Information

## Acknowledgements

MA Sussman is a recipient of funding from the California Tobacco Related Disease Research Program (Pilot award T31IP1790 and Impact award T31IR1585). The author extends his deepest appreciation to members of the Sussman Laboratory who provide invaluable assistance and expertise in developing vaping-related studies and information. This publication includes data generated at the UC San Diego IGM Genomics Center utilizing an Illumina NovaSeq 6000 that was purchased with funding from a National Institutes of Health SIG grant (#S10 OD026929)." Apologies to all authors and researchers in this field of study for their relevant citations that were not included in this review due to space constraints as well as the substantial amount of excellent published material in this field.

### Author Contributions

C Esquer: data curation and investigation.
O Echeagaray: formal analysis, investigation, and writing—original draft, review, and editing.
F Firouzi: formal analysis, investigation, and writing—original draft, review, and editing.
C Savko: data curation, formal analysis, investigation, methodology, and writing—original draft, review, and editing.
G Shain: data curation, formal analysis, investigation, methodology, and writing—original draft, review, and editing.
P Bose: investigation.
A Rieder: investigation.
S Rokaw: investigation.
A Witon-Paulo: investigation.
N Gude: investigation.
MA Sussman: conceptualization, formal analysis, supervision, funding acquisition, validation, investigation, methodology, project administration, and writing—original draft, review, and editing.

### Conflict of Interest Statement

The authors declare that they have no conflict of interest.

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
