## [Reviewer comments · Life Science Alliance]

Life Science Alliance

Fundamentals of Vaping-Associated Pulmonary Injury Leading to Severe Respiratory Distress

Carolina Esquer, Oscar Echeagaray, Fareheh Firouzi, Clarissa Savko, Grant Shain, Pria Bose, Abigail Rieder, Sophie Rokaw, Andrea Witon-Paulo, Natalie Gude, and Mark Sussman

DOI: <https://doi.org/10.26508/lsa.202101246>

Corresponding author(s): Mark Sussman, San Diego State University

Review Timeline:	Submission Date:	2021-09-24
	Editorial Decision:	2021-10-21
	Revision Received:	2021-10-30
	Editorial Decision:	2021-11-01
	Revision Received:	2021-11-04
	Accepted:	2021-11-05

Transaction Report:

October 21, 2021

Re: Life Science Alliance manuscript #LSA-2021-01246-T

Prof. Mark Alan Sussman
San Diego State University
Biology
5500 Campanile Drive
San Diego, CA 92129

Dear Dr. Sussman,

Thank you for submitting your manuscript entitled "Fundamentals of Vaping-Associated Pulmonary Injury Leading to Severe Respiratory Distress" to Life Science Alliance. The manuscript was assessed by expert reviewers, whose comments are appended to this letter. We invite you to submit a revised manuscript addressing the Reviewer comments.

Thank you for this interesting contribution to Life Science Alliance. We are looking forward to receiving your revised manuscript.

Sincerely,

B. MANUSCRIPT ORGANIZATION AND FORMATTING:

Reviewer #1 (Comments to the Authors (Required)):

The study by Carolina Esquer et al. deals with the very timely and relevant issue of lung injury by vaping. It has been indeed well demonstrated that vaping products can cause acute, severe respiratory distress. The latter represents a significant concern considering the widespread adoption of vaping as a social activity and lifestyle choice by 'never smokers', particularly adolescents. The authors should be congratulated for creating a reproducible and effective mouse model of lung injury by vaping that has the particular merit of closely resembling human scenario. Indeed, this study demonstrates for the first time Vaping-Induced Pulmonary Injury (VAPI) using commercial JUUL pens with flavored vape juice using an inhalation exposure murine model. Profound pathological changes to upper airway, lung tissue architecture, and cellular structure are evident within 9 weeks of exposure. Increased parenchyma tissue density, cellular infiltrates proximal to airway passages, alveolar rarefaction, increased collagen deposition, and bronchial thickening with elastin fiber disruption were changes detected by histological analysis. Also the lung tissue was analyzed by transcriptome analysis that revealed transcriptional reprogramming including significant changes to gene families coding for xenobiotic response, glycerolipid metabolic processes, and oxidative stress. As final test of the reliability of the acute injury model, the authors show that cardiac contractile performance for systemic output is moderately but significantly impaired. Overall, this VAPI model with pulmonary circuit failure demonstrates mechanistic underpinnings of vaping-related pathologic injury and is set to be an invaluable model for basic and pre-clinical research to clearly inform about the risk of vaping and how to prevent it.

I strongly support the publication of this study as indeed the quality of the data is very high and the model is very deeply characterized. I have only few suggestions to the authors:

1. A few times the authors write that is not yet known 'why' do particular individuals develop VAPI; I would leave the why to psychiatrists and keep the 'how' to basic science.
2. The introductions and discussion are in few instances repetitive and quite long. I would therefore suggest shortening and strengthening them both.

Reviewer #2 (Comments to the Authors (Required)):

This paper reports the phenotypic characterization of vaping-induced pulmonary injury and cardiac remodeling in a unique mouse model. The topic is certainly timely, the paper is well written, and the analyses are largely well performed and described. This is an impressive amount of work. As my expertise lies outside of that in pulmonary injury associated with vaping or other environmental factors, I limit my critique to suggestions for improvement on the technical aspects of the manuscript. I've suggested this as a "major revision" but believe that the critiques suggested could be addressed using existing data and in a relatively short period of time.

Before critiquing the results of Figures 1-5, I'd like to commend the authors on some really outstanding microscopy.

1. The representation of Figure 1E could use some clarification. Is this a box+whiskers plus outliers? Is each measurement considered as an independent observation? Since measurements in each animal are linked, considering each as an independent replicate would be inappropriate. Using the median value per mouse would be a more appropriate value for each replicate.
2. Figure 2-5: Ideally, the elastin, collagen, and mucin staining-based analyses would be accompanied by an attempt at quantification before claims of increased deposition or increase disorganization. The Western in 3C is a solid method that clarifies the mucin accumulation. Perhaps quantification of methyl blue stain in the trichrome would be helpful. Elastin may prove more challenging, in which case the statements should likely be weakened. Along the same line, although proteins associated with infiltrate are quantified by Western in Figure 5, one could also quantify the number of cells per unit area that are present. It is likely that the data to perform all of the above suggested analysis already exist.

Spatial Transcriptomics

I'd applaud the authors for moving into this technology with enthusiasm. These approaches can give unprecedented insight into transcriptional effects in anatomical dimensions that are otherwise unattainable. The analyses associated with these are non-trivial and constantly changing, so I offer a few suggestions to improve the analysis, particularly as it pertains to the differential

expression portion.

1. General comment: Low spatial resolution technologies such as Visium are poorly suited for highly admixed tissues such as the lung. The authors correctly recognize this and do their downstream differential expression analysis in a smart way; treating it as a regionally restricted bulk-RNAseq experiment. The deconvolution of the spots into their respective cell types using a reference dataset in order to do compositional comparisons is a great analysis. Going forward, I'd love to see this analysis performed in space irrespective of cluster (think of a higher resolution version of panel 6D analyzed for composition as you move through the tissue), but this is unnecessary for the acceptance of the manuscript.

2. Some of the analytic methodology for the ST work is very terse. While the upstream processing appears to be rather standard, the model for DE appears to be problematic. First, was the approach performed on a total of four mice, with one male and one female per set? Single cell (or single spot, in this case) data have intraindividual correlation, artificially reducing the observed variability if one treats each spot or cell as an individual observation (as it is in the Wilcoxon rank-sum analysis). Using a framework that accounts for this, such as the now field standard MAST workflow, would give more appropriate DE results. This would inevitably reduce the number of DE genes, but even if the majority are subsignificant, these results can be used in a GSEA analysis to look at pathway enrichment. Since the data associated with ST work is much more sparse than that typically found in dissociative approaches, it is likely that this work could be completed rather quickly on a personal computer. It was also unclear how the DE results were corrected for multiple testing, B-H FDR method?

Reviewer #3 (Comments to the Authors (Required)):

In their manuscript, Esquer and colleagues present a timely study of a mouse model of vaping. They utilize the most relevant (though not all-encompassing) device available and expose mice to ecig vapor for 9 weeks and examine the lungs and heart. The presentation of data is very impressive, and there are only a few problems in this very timely manuscript.

The authors find that vaping induces (generally) 1) pulmonary structural alterations demonstrating lung injury 2) pulmonary inflammation and 3) mild cardiovascular remodeling. The authors also provide a spacial transcriptional dataset that demonstrates clear mechanisms underlying the pulmonary damage. The data are well-presented and include all needed information.

A few minor thoughts:

Please comment on the age of the mice upon exposure (6-8 weeks) and any limitations of exposing adult mice vs. juvenile mice, the latter perhaps modeling the main Ecig user base.

What are the long-term implications of the damages found? Would the mice (an most importantly the human user) be expected to recover after quitting?

Department of Biology
College of Sciences
San Diego State University
5500 Campanile Drive
San Diego CA 92182 · 4614
Tel: 619 · 594 · 6767
Fax: 619 · 594 · 5676

10/29/2021
Eric Sawey, PhD
Executive Editor
Life Science Alliance
<http://www.lsjournal.org>

Dear Dr. Sawey,

We thank the reviewers for their supportive and valuable feedback to improve upon our original submission entitled "Fundamentals of Vaping-Associated Pulmonary Injury Leading to Severe Respiratory Distress" (#LSA-2021-01246-T). Detailed responses and revisions to the original manuscript are provided below *in italics*. We look forward to the comments from reviewers on our revision and hope that the manuscript is now suitable for publication in *Life Science Alliance*.

Reviewer #1:

The study by Carolina Esquer et al. deals with the very timely and relevant issue of lung injury by vaping...The authors should be congratulated for creating a reproducible and effective mouse model of lung injury by vaping that has the particular merit of closely resembling human scenario. Indeed, this study demonstrates for the first time Vaping-Induced Pulmonary Injury (VAPI) using commercial JUUL pens with flavored vape juice using an inhalation exposure murine model...I strongly support the publication of this study as indeed the quality of the data is very high and the model is very deeply characterized.

We truly appreciate the enthusiastic endorsement of our study by the reviewer.

I have only few suggestions to the authors:

1) A few times the authors write that is not yet known 'why' do particular individuals develop VAPI; I would leave the why to psychiatrists and keep the 'how' to basic science.

Point is well taken and we have revised the text accordingly to reflect focus upon biological perspective (Page 4, line 53 and Page 20, line 430).

2) The introduction and discussion are in few instances repetitive and quite long. I would therefore suggest shortening and strengthening them both.

Text for the Introduction and Discussion sections has been reassessed and trimmed to eliminate redundancies as suggested.

Reviewer #2:

This paper reports the phenotypic characterization of vaping-induced pulmonary injury and cardiac remodeling in a unique mouse model. The topic is certainly timely, the paper is well written, and the analyses are largely well performed and described. This is an impressive amount of work.

We sincerely thank the reviewer for these supportive and encouraging comments on our study.

1) Before critiquing the results of Figures 1-5, I'd like to commend the authors on some really outstanding microscopy. The representation of Figure1E could use some clarification. Is this a box+whiskers plus outliers? Is each measurement considered as in independent observation? Since measurements in each animal are linked, considering each as an independent replicate would be inappropriate. Using the median value per mouse would be a more appropriate value for each replicate.

The graph for Figure 1E represented all independent measurements with the average plotted as a horizontal bar. As the reviewer correctly asserts, measurements within an animal are linked so the original graph has been replaced with an updated version. In the revised version each dot represents the average of all measurements within one animal and the bars representing the median and interquartile range. The figure legend (Page 37 Line 954-956) and Methods (Page 8 Line 144) have been edited accordingly to reflect these updates.

2) Figure 2-5: Ideally, the elastin, collagen, and mucin staining-based analyses would be accompanied by an attempt at quantification before claims of increased deposition or increase disorganization. The Western in 3C is a solid method that clarifies the mucin accumulation. Perhaps quantification of methyl blue stain in the trichrome would be helpful. Elastin may prove more challenging, in which case the statements should likely be weakened. Along the same line, although proteins associated with infiltrate are quantified by Western in Figure 5, one could also quantify the number of cells per unit area that are present. It is likely that the data to perform all of the above suggested analysis already exist.

We appreciate the reviewer's perspective on quantitative analyses of the microscopy. As suggested we performed quantification of methyl blue on the trichrome-stained sections using ImageJ. Quantifying the images from Fig 2 of the manuscript revealed 73.12% (1.73-fold) increase of collagen deposition in the vaped lung compared to the non-vaped group (non-vaped covered 3.18 % of Area; vaped covered 5.51 % of Area). A second quantitation performed upon an unrelated set of vaped and non-vaped lung tile scans yielded a 22.37% (1.23-fold) increase of collagen deposition in the vaped lung compared to the non-vaped group (non-vaped covered 3.7 % of Area; vaped covered 4.6 % of Area). Taken together, this n=2 sampling results in an average increase of 45.69% (1.5-fold) increase for collagen deposition in the vaped lung compared to the non-vaped group (non-vaped covered 3.46 % of Area; vaped covered 5.04 % of Area). The text of the results has been updated accordingly with these additional measurements resulting from averaging the two separate samples together (Page 13, lines 244-246).

On the second suggestion of elastin measurement we concur that quantitation of "disorganization" is challenging and have revised the text to state that elastin organization is "alteration" in vaped samples compared to non-vaped controls (Page 12, line 240).

Lastly on the topic of quantitation of cellular infiltrate, comparison of cellular density for CD11b and CD11c was determined by counting of cells in sections from two non-vaped and four vaped mouse samples. Four images were taken per sample, each with an area of 1.32 mm² totaling 5.27 mm² imaged per sample. Cell count for CD11B was significantly increased by four fold (8.875 ± 1.619 for no vape versus 35.69 ± 4.654 vape; $p=0.001$). Cell count for CD11C was significantly increased by 2.3 fold (23.25 ± 7.2 no vape versus 53.19 ± 5.257 vape; $p=0.003$). This information has been incorporated into the Results section text (Page 15, line 292-297).

Spatial Transcriptomics. I'd applaud the authors for moving into this technology with enthusiasm. These approaches can give unprecedented insight into transcriptional effects in anatomical dimensions that are otherwise unattainable. The analyses associated with these are non-trivial and constantly changing, so I offer a few suggestions to improve the analysis, particularly as it pertains to the differential expression portion.

1) General comment: Low spatial resolution technologies such as Visium are poorly suited for highly admixed tissues such as the lung. The authors correctly recognize this and do their downstream differential expression analysis in a smart way; treating it as a regionally restricted bulk-RNAseq experiment. The deconvolution of the spots into their respective cell types using a reference dataset in order to do compositional comparisons is a great analysis. Going forward, I'd love to see this analysis performed in space irrespective of cluster (think of a higher resolution version of panel 6D analyzed for composition as you move through the tissue), but this is unnecessary for the acceptance of the manuscript.

We appreciate the reviewer's enthusiasm for the effort put forth in our study as well as suggestions to elevate the quality of our data. Deconvolution of the spots transcriptome was done irrespective of cluster. Given the number of spots in the analysis (4933), we considered the representation by cluster as presented in Figure 6 and Supplemental Figures 10 and 11 to be the most effective way to visually cross-reference

deconvolution and spatial coordinates within the tissue. Following the reviewer's suggestion, the correlation scores per spot derived from the deconvolution to the Mouse Cell Atlas have been included in Supplemental File 1 and mentioned on the text (Page 16 Lines 326-327) as follows:

"Spot mapping identified cell types in each cluster by score, with some spots mapping to multiple cell types (SF8 and Supplemental table 1)."

2) Some of the analytic methodology for the ST work is very terse. While the upstream processing appears to be rather standard, the model for DE appears to be problematic. First, was the approach performed on a total of four mice, with one male and one female per set? Single cell (or single spot, in this case) data have intraindividual correlation, artificially reducing the observed variability if one treats each spot or cell as an individual observation (as it is in the Wilcoxon rank-sum analysis). Using a framework that accounts for this, such as the now field standard MAST workflow, would give more appropriate DE results. This would inevitably reduce the number of DE genes, but even if the majority are subsignificant, these results can be used in a GSEA analysis to look at pathway enrichment. Since the data associated with ST work is much more sparse than that typically found in dissociative approaches, it is likely that this work could be completed rather quickly on a personal computer. It was also unclear how the DE results were corrected for multiple testing, B-H FDR method?

We appreciate the reviewer's input on our analysis. Part of our analytical strategy included selection of a differential gene expression test suited to our dataset. Based upon the spatial transcriptome dataset with respect to previously published comparative analysis (Wang et al., 2019), we originally selected Model-based Analysis of Single-cell Transcriptomics (MAST) test as the reviewer suggests. However, MAST DE test failed to identify DEGs from our datasets. We interpret this outcome to reflect that the MAST test was designed with consideration toward the multimodality and sparsity of single cell data (Wang et al., 2019). However, applicability of the MAST test to spatial transcriptomic data relative to the features and assumptions used for single cell data are unclear and possibly inappropriate. We suspect the multimodality and sparsity of spatial transcriptomic data from highly admixed tissues as the lung, which holds the transcriptome of numerous mixed cells within a 50 μm spot does not compare to single cell transcriptome derived from a microfluidics approach. Of course, the number of samples processed per Visium slide ($n=4$, two samples per group, one male and one female per group) is a technical issue that may also limit interpretation by MAST test. However, follow up on expression levels per spot and spots count of targets of interest were robust and consistent with Wilcoxon test results (Fig 7D-7D, 8D, 8G and SF 14 and 15D). Therefore, we believe these DEG results are valid and appropriate. To highlight these considerations we have incorporated the following text into the Discussion (Page 25, lines 546-555):

"The differential expression analysis strategy of Model-based Analysis of Single-cell Transcriptomics (MAST) test (Wang et al., 2019) did not yield DEGs in our analysis. However, applicability of the MAST test to spatial transcriptomic data relative to the features and assumptions used for single cell data are unclear and possibly inappropriate. We suspect the multimodality and sparsity of spatial transcriptomic data from highly admixed tissues as the lung, which holds the transcriptome of numerous mixed cells within a 50 μm spot does not compare to single cell transcriptome derived from a microfluidics approach. Nevertheless, DEG results are valid based upon expression levels per spot and spots count of targets of interest that were robust and consistent with Wilcoxon test results (Fig 7D-7D, 8D, 8G and SF 14 and 15D)."

We also included in the methods the multiple hypothesis test methods to control the number of false positives (Page 9 Line 182-183) as follows: "Bonferroni correction was use as a multiple hypothesis test method to control the number of false positives (Diz et al., 2011)."

References

Diz, A. P., Carvajal-Rodríguez, A., & Skibinski, D. O. F. (2011). Multiple Hypothesis Testing in Proteomics: A Strategy for Experimental Work. *Molecular & Cellular Proteomics*, 10(3), M110.004374.

Wang, T., Li, B., Nelson, C. E., & Nabavi, S. (2019). Comparative analysis of differential gene expression analysis tools for single-cell RNA sequencing data. *BMC Bioinformatics* 2019 20:1, 20(1), 1–16.

Reviewer #3

In their manuscript, Esquer and colleagues present a timely study of a mouse model of vaping. They utilize the most relevant (though not all-encompassing) device available and expose mice to ecig vapor for 9 weeks and examine the lungs and heart. The presentation of data is very impressive, and there are only a few problems in this very timely manuscript. The data are well-presented and include all needed information.

We thank the reviewer for the positive feedback on our study and constructive suggestions.

1) Please comment on the age of the mice upon exposure (6-8 weeks) and any limitations of exposing adult mice vs. juvenile mice, the latter perhaps modeling the main Ecig user base.

The reviewer highlights a very important consideration that was indeed a major consideration in our experimental design. During the adult phase of mouse life, 2.6 days are approximately equivalent to one human year. We vape our mice starting at approximately 8 weeks (56 days) in a time course of 8-9 weeks (age at conclusion equals 112 – 119 days). This represents the equivalent of human vaping from 16 years to 37.5 – 40.2 years of age according to Dutta and Sengupta (2016) or the equivalent of 18 – 25 years of age according to Flurkey (2007). This information has been added to the Methods section (Page 7, lines 112-115).

References

Dutta S, and Sengupta P. Men and mice: Relating their ages. Life Sci. 2016;152:244-8.

Flurkey K, M. Currer J, and Harrison DE. In: Fox JG, Davisson MT, Quimby FW, Barthold SW, Newcomer CE, and Smith AL eds. The Mouse in Biomedical Research (Second Edition). Burlington: Academic Press; 2007:637-72.

2) What are the long-term implications of the damages found? Would the mice (and most importantly the human user) be expected to recover after quitting?

This is a very important question and we certainly look forward to evaluating recovery from VAPI in future studies using our model. Damage from VAPI may exhibit distinctive severity including features such as lipid pneumonia and also involve features specific to particular vaping agents such as cinnamaldehyde or nicotine. The diverse array of vaping behaviors and products in use presents a problem for predictions of recovery from VAPI depending upon underlying pathological drivers and individual characteristics of afflicted patients. These are early days for tracking patient long term recovery outcomes that can take several years to be fully realized. For all these reasons, I chose to not digress into the issue of long term VAPI recovery in this submission. This topic was touched upon in my recently published review article that I have now incorporated that into the Discussion as follows (Page 27, line 605-607):

“Only the passage of time will provide the information needed to assess the long term consequences of VAPI and recovery potential for human vapers, but these are certainly areas worthy of further investigation. (Sussman MA, 2021).”

Reference

Sussman MA. VAPIng into ARDS: Acute Respiratory Distress Syndrome and Cardiopulmonary Failure. Pharmacol Ther. 2021 Sep 25:108006. Epub ahead of print. PMID: 34582836.

November 1, 2021

RE: Life Science Alliance Manuscript #LSA-2021-01246-TR

Prof. Mark Alan Sussman
San Diego State University
Biology
5500 Campanile Drive
San Diego, CA 92129

Dear Dr. Sussman,

Thank you for submitting your revised manuscript entitled "Fundamentals of Vaping-Associated Pulmonary Injury Leading to Severe Respiratory Distress". We would be happy to publish your paper in Life Science Alliance pending final revisions necessary to meet our formatting guidelines.

- Please upload all figure files individually, including the supplementary figure files
- please consult our manuscript preparation guidelines <https://www.life-science-alliance.org/manuscript-prep> and make sure your manuscript sections are in the correct order
- please add an Author Contributions section to your main manuscript text
- please add a conflict of interest statement to your main manuscript text
- please add your main and supplementary figure legends to the main manuscript text after the references section
- please upload your Tables in editable .doc or excel
- please use capital letters when labeling panels in figures and their legends
- please add callouts for Figures 5D, 8D, 10A-D, S2A-C, S7A-D, S8A-G, S9A-B, S14A-C, S15A-D, and S16A-B to your main manuscript text
- Please indicate molecular weight next to each protein blot
- Please incorporate the methods provided in the Supplemental file into the main manuscript file. We do not have any limit on the size of the Materials and Methods section.

A. FINAL FILES:

B. MANUSCRIPT ORGANIZATION AND FORMATTING:

Sincerely,

Reviewer #2 (Comments to the Authors (Required)):

All of my concerns have been addressed appropriately. Well done!

Reviewer #3 (Comments to the Authors (Required)):

My previous comments have successfully been addressed.

November 5, 2021

RE: Life Science Alliance Manuscript #LSA-2021-01246-TRR

Prof. Mark Alan Sussman
San Diego State University
Biology
5500 Campanile Drive
San Diego, CA 92129

Dear Dr. Sussman,

Thank you for submitting your Research Article entitled "Fundamentals of Vaping-Associated Pulmonary Injury Leading to Severe Respiratory Distress". It is a pleasure to let you know that your manuscript is now accepted for publication in Life Science Alliance. Congratulations on this interesting work.

DISTRIBUTION OF MATERIALS:

Again, congratulations on a very nice paper. I hope you found the review process to be constructive and are pleased with how the manuscript was handled editorially. We look forward to future exciting submissions from your lab.

Sincerely,
